# Genetic architecture of plasma metabolome in 254,825 individuals

Yi-Xuan Qiang[1,5], Yi-Xuan Wang[1,5], Xiao-Yu He[1,5], Yue-Ting Deng [1,5], Yi-Jun Ge [1,5], Bang-Sheng Wu[1], You Jia [1,2,3], Jian-Feng Feng [2,3,4], Wei Cheng [1,2,3] ✉ & Jin-Tai Yu [1] ✉

Circulating metabolites are crucial to biological processes underlying health and diseases, yet their genetic determinants remain incompletely understood. Here, we investigate the genetic architecture of nuclear magnetic resonance-based metabolomics, analyzing 249 metabolic measures and 64 biologically plausible ratios in 254,825 participants. We conduct a genome-wide association study (GWAS) identifying 24,438 independent variant-metabolite associations across 427 loci, with effect sizes highly concordant with 19 previous studies. Fine-mapping pinpoints 3610 putative causal associations, 785 of which are novel. Additionally, we utilize whole exome sequencing data and uncover 2948 gene-metabolite associations through aggregate testing, underscoring the importance of rare coding variants overlooked in GWAS. Integrating our findings with disease genetics reveals potential causal associations, such as between acetate levels and the risk of atrial fibrillation and flutter. Collectively, this study delineates the complex genetic architecture of the plasma metabolome, offering a valuable resource for future investigations into disease mechanisms and therapeutic strategies.

Circulating metabolites serve as crucial indicators of cellular processes and tissue function[1,2], with their abnormalities being a prevalent characteristic across various health conditions. Many plasma metabolites exhibit high levels of heritability, suggesting a strong genetic component in their regulation[3]. However, only a fraction of the underlying genetic variants has been identified to date. By integrating metabolomics and genomics data in a population-based cohort, it is possible to better characterize the still-missing genetic determinants, gain insights into their involvement in disease mechanisms, and identify molecular targets for therapeutic interventions.

Numerous studies have explored the genetic architecture of the human metabolome using diverse cohorts and analytical platforms. However, most studies faced limitations regarding either sample size or metabolite diversity[4–9]. A recent study involving 136,016 individuals

from 33 cohorts identified 212 novel loci, highlighting the potential for further discoveries in larger cohorts[10]. While this multi-cohort study provided valuable insights, it encompassed inherent heterogeneity in participant characteristics, sample types, and fasting statuses. Furthermore, metabolite ratios can serve as sensitive biomarkers by reducing external variability and reflecting pathway functionality[11]. Nuclear magnetic resonance (NMR)-based metabolite ratios offer even greater clinical application value due to their cost-effectiveness[12]. However, previous studies examining metabolite ratios have predominantly focused on mass spectrometry (MS)-based profiles or employed hypothesis-free approaches, with sample sizes typically below 10,000 individuals[13–18]. Additionally, while genetic studies of the metabolome have mostly focused on common variants, incorporating rare coding variants can provide novel insights[19–22]. Therefore, there is

[1]Department of Neurology and National Center for Neurological Disorders, Huashan Hospital, State Key Laboratory of Medical Neurobiology and MOE Frontiers Center for Brain Science, Shanghai Medical College, Fudan University, Shanghai, China. [2]Institute of Science and Technology for Brain-Inspired Intelligence, Fudan University, Shanghai, China. [3]Key Laboratory of Computational Neuroscience and Brain-Inspired Intelligence, Fudan University, Ministry of Education, Shanghai, China. [4]Department of Computer Science, University of Warwick, Coventry, UK. [5]These authors contributed equally: Yi-Xuan Qiang, Yi-Xuan Wang, Xiao-Yu He, Yue-Ting Deng, Yi-Jun Ge. ✉e-mail: wcheng@fudan.edu.cn; jintai_yu@fudan.edu.cn

a pressing need for large-scale research that incorporates both common and rare variants to thoroughly investigate NMR metabolites and their associated ratios.

In this study, we comprehensively investigate the genetic architecture of plasma NMR-based metabolomics, analyzing 249 metabolic measures and 64 biologically relevant ratios in a cohort of 254,825 participants from the UK Biobank (UKB)[23]. We identify and refine common variants (minor allele frequency [MAF] ≥ 1%) associated with metabolic traits through genome-wide association study (GWAS), clumping, and fine-mapping. These findings were further validated using results from 19 metabolomics GWAS. The genetic architecture of metabolites underwent comprehensive investigation, including evaluations of polygenicity, pleiotropy, heritability, and genetics-phenotype concordance. To explore the contribution of rare variants (MAF < 1%) to the NMR metabolome, we further leveraged whole exome sequencing (WES) data to perform exome-wide gene-based collapsing analyses. Finally, colocalization and Mendelian randomization (MR) analyses are conducted to assess the shared genetic determinants and potential causal relationships between metabolites and diseases.

## Results

### Population characteristics
We included 254,825 UKB participants with both metabolomic and genomic data available after quality control. The median age was 58 years (interquartile range, IQR: 50–63 years), with 137,089 (53.79%) individuals being female and 218,380 (85.70%) individuals of white British ancestry. The median fasting time before blood collection was 3 h (IQR: 2–4 h). The study workflow is illustrated in Fig. 1, and the characteristics of the study population are delineated in Supplementary Data 1.

### GWAS of 249 plasma metabolic measures and 64 derived ratios
The primary GWAS analysis included 189,846 white British participants and 7,924,871 common variants (MAF ≥ 1%) after quality control. Plasma levels of 249 metabolic measures were quantified utilizing a high-throughput NMR metabolic biomarker platform developed by Nightingale Health Ltd[24]. A substantial proportion of the identified metabolites were grouped into the lipoprotein and lipid category (n = 192; 77.11%), with the remaining metabolites classified as amino acids (n = 10; 4.02%), apolipoproteins (n = 3; 1.20%), cholesterol (n = 15; 6.02%), fatty acids (n = 18; 7.23%), fluid balance (n = 2; 0.80%), glycolysis-related (n = 4; 1.61%), inflammation (n = 1; 0.40%), and ketone bodies (n = 4; 1.61%) (Supplementary Data 2). The primary GWAS analysis of 249 metabolic measures identified 21,132 independent variant-metabolite associations at 3059 unique lead variants (Methods). These associations were distributed across the entire NMR metabolomic spectrum, with the majority falling into the lipoprotein and lipid category, where the number of associations per trait ranged from 24 to 152. Fewer associations were observed for non-lipid traits, ranging from 7 to 85 per trait (Fig. 2, Supplementary Data 3 and 4).

We also analyzed ratios of 64 metabolite pairs linked by common proteins (Fig. 3), as documented in the Human Metabolome Database (HMDB). Among these ratios, about half were within the amino acid group (n = 33; 51.56%), and approximately a quarter involved metabolites related to glycolysis (n = 17; 26.56%) (Supplementary Data 5). We identified 3306 independent associations for the 64 metabolite ratios. The number of associations per metabolite ratio ranged from 6 (isoleucine to valine ratio) to 122 (total cholesterol to total triglycerides ratio) (Supplementary Data 4). Across the 64 ratios, ratio-associated loci showed comparable overlap with both denominator-associated (0%–68.18%) and numerator-associated loci (0%–77.78%) (Supplementary Data 6). The median proportion of lead variants that showed stronger associations with the ratio than with either individual metabolite was 33.77% (IQR: 12.74%–53.04%), based on a P-gain threshold of

640. Notably, a median of 21.26% (IQR: 9.53%–57.89%) of loci were uniquely identified through ratio-based analyses, underscoring their added value in uncovering novel associations beyond individual metabolites (Supplementary Data 7).

The linkage disequilibrium (LD) score regression intercepts[25] for all 313 metabolic traits ranged from 0.99 to 1.09, with a median value of 1.04. These intercepts indicate a reasonable inflation (not substantially above 1), which can predominantly be attributed to the polygenicity of the metabolic traits[4]. Each independent variant explained from 0.02% to 11.13% (median = 0.04%) of the phenotypic variance of the corresponding trait, with 266 (1.09%) variants explaining ≥ 1% of the phenotypic variance (Supplementary Fig. 1).

Subgroup analyses stratified by sex revealed that 13,770 out of 24,438 lead variant–trait associations were significant in both male and female subgroups ($P < 1 \times 10^{-6}$; Supplementary Data 4). Sex-specific associations were identified, exemplified by rs768832539, which was significantly associated with glycine levels in females ($P = 3.77 \times 10^{-40}$) but not in males ($P = 0.16$). Interestingly, this variant has also been reported to show female-specific effects on sleep duration according to the GWAS Atlas[26,27]. These findings suggest that sex-specific metabolic associations may provide mechanistic insights, such as potential mediating pathways, into other important human phenotypes.

### Pleiotropy, polygenicity, and heritability of metabolic measures and derived ratios
The GWAS analysis elucidated the complex genetic architecture underlying metabolites and derived ratios. The number of associations identified in each metabolite category was proportional to the number of metabolites within that category (Fig. 4a). We then assessed single-locus pleiotropy across 427 independent loci, derived from 24,438 lead variant-trait associations. The majority of these loci were associated with multiple traits (n = 323; 75.64%). Specifically, 37 loci (8.67%) were linked only to traits within the same category (intra-categorical pleiotropy) and 286 loci (66.98%) were associated with traits from different categories (inter-categorical pleiotropy) (Figs. 2, 4b and Fig. 4b, Supplementary Data 8). Notably, *TRIB1* (8q24.13) exhibited the most extensive inter-categorical pleiotropy, being associated with 255 traits across 9 categories. Our results also bolster the concept of metabolites' polygenicity. We observed that each trait was associated with a minimum of 5 independent loci, with 171 traits associated with over 50 loci. Certain metabolites, such as cholesteryl esters in large high-density lipoprotein (HDL), were associated with up to 85 loci (Fig. 2).

The median estimated heritability across all 313 metabolic traits was 12.32%, with significant variability among different categories. Specifically, heritability was higher in the categories of lipoprotein and lipid (14.33%), fatty acids (13.18%), and apolipoproteins (12.85%), while glycolysis-related metabolites (5.76%) and ketone bodies (3.29%) exhibited lower heritability (Fig. 4c, Supplementary Data 9). Notably, a positive correlation was observed between the number of associated genetic loci and the heritability estimates of metabolites (Supplementary Fig. 2). Subsequent subgroup analysis revealed that four out of the ten categories showed significant associations ($P < 0.05$), while the lack of significance in the remaining six categories may stem from the limited number of metabolites within these categories (Fig. 4d). Additionally, we examined the pairwise phenotypic and genetic correlations among the 313 metabolic traits, revealing a highly concordant structure between phenotypic and genetic correlation matrices (Supplementary Fig. 3). Together, these findings highlight the complex genetic architecture of metabolic traits, characterized by polygenicity, pleiotropy, and a strong concordance between genetics and phenotypes.

### Fine mapping reveals candidate causal variants
We next performed fine-mapping analysis using the FINEMAP software[28] to prioritize genetic variants most likely to have a direct

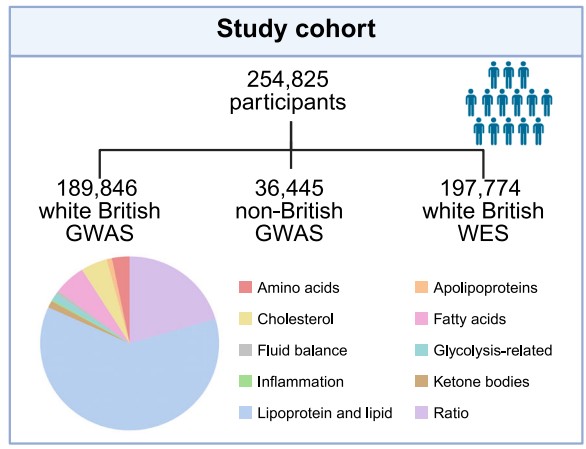

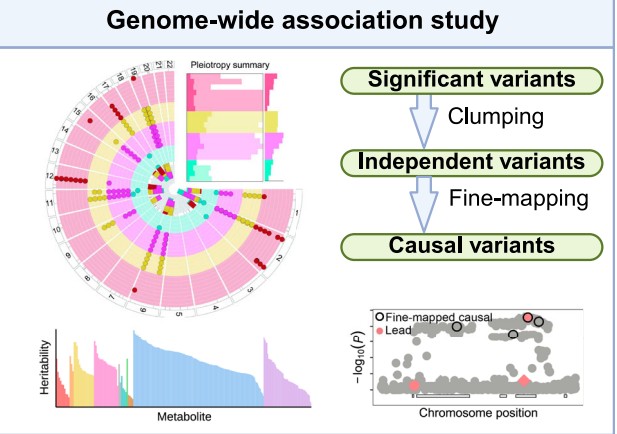

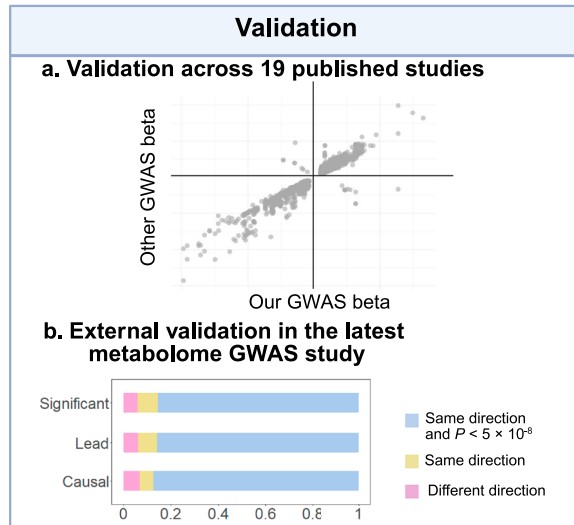

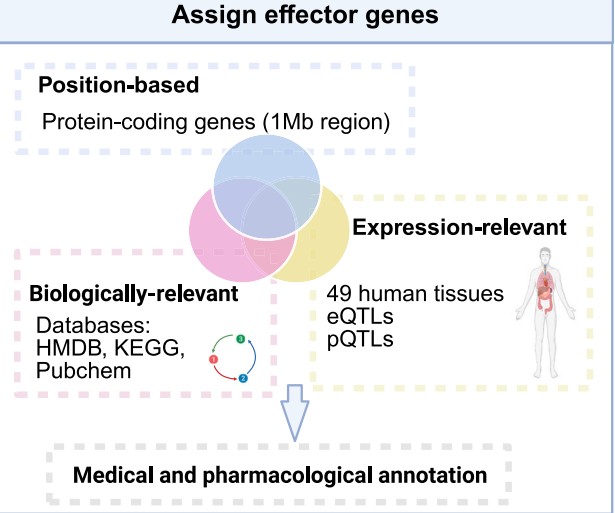

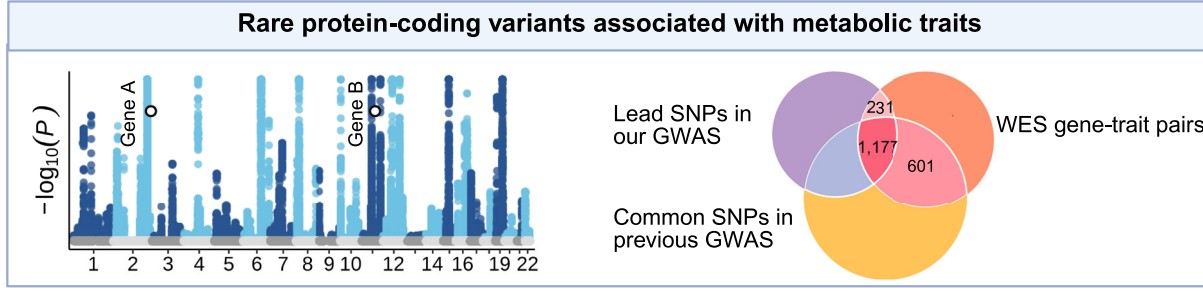

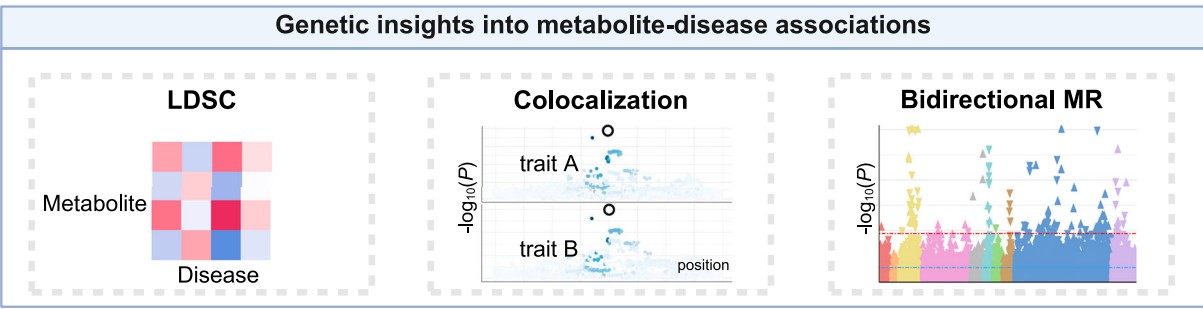

**Fig. 1 | Graphical summary of the study workflow and results.** Created in BioRender. Qiang, Y. (2025) https://BioRender.com/b5b9nh8. Abbreviations: GWAS genome-wide association study, HMDB the Human Metabolome Database, KEGG the Kyoto Encyclopedia of Genes and Genomes Pathway database, LDSC linkage disequilibrium score regression, MR Mendelian randomization, WES whole exome sequencing.

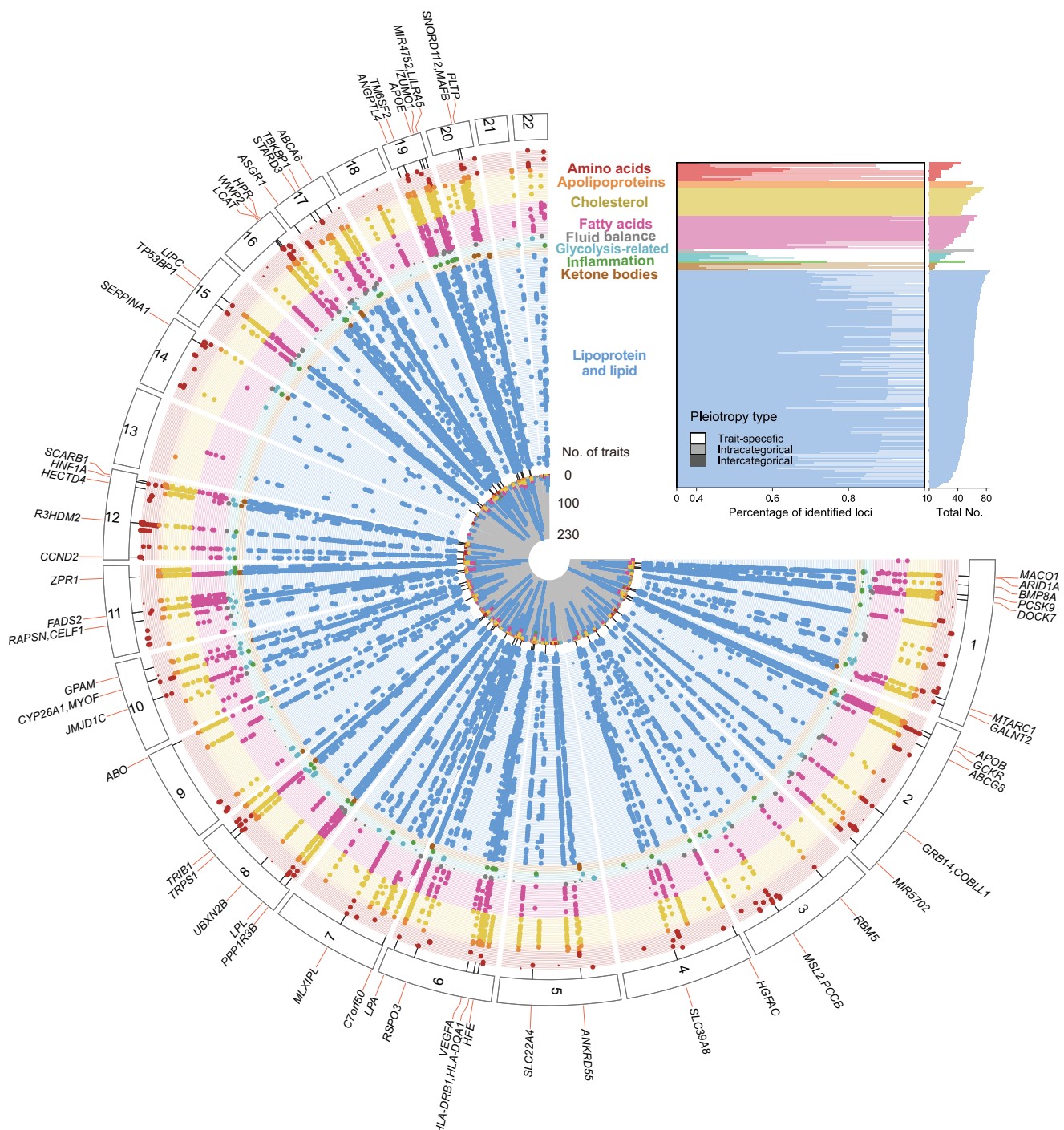

**Fig. 2 | Overview of the identified loci associated with 249 metabolic measures and their pleiotropy effects.** Top right inset, the number of trait-associated loci for each metabolite, grouped by their pleiotropy type. Main panel, Fuji plot illustrating the GWAS results of 249 metabolic measures across 9 categories. Each dot represents a locus-trait association, with larger dots indicating pleiotropic associations. Each radial line connects all dots for an intercategorical pleiotropic locus, which are associated with traits from more than four categories, with a locus symbol. Source data are available in Supplementary Data 8.

effect on metabolic traits[29,30]. We created 2-Mb regions centered on each index variant and merged any overlapping regions associated with the same metabolite. Among the 14,661 unique regions, we identified 35,338 distinct signals. In the identified credible sets, 13,597 signals were fine-mapped to 5 or fewer variants with a posterior probability of including the causal variant $P > 0.99$; at 6620 trait-associated loci, we resolved the signal to a single nominated causal variant (Fig. 5a and Supplementary Data 10). We identified

3610 unique variant–metabolite associations with a posterior probability $P > 0.99$ for being the causal variant. Overall, the putative causal variants demonstrated an inverse relationship between their effect allele frequency and absolute effect size (Supplementary Fig. 4). Of these, 2871 overlap with the 24,438 independent associations identified through clumping analysis. Notably, cholesteryl esters in large HDL and triglycerides to total lipids in very small very low-density lipoprotein (VLDL) percentage had the most

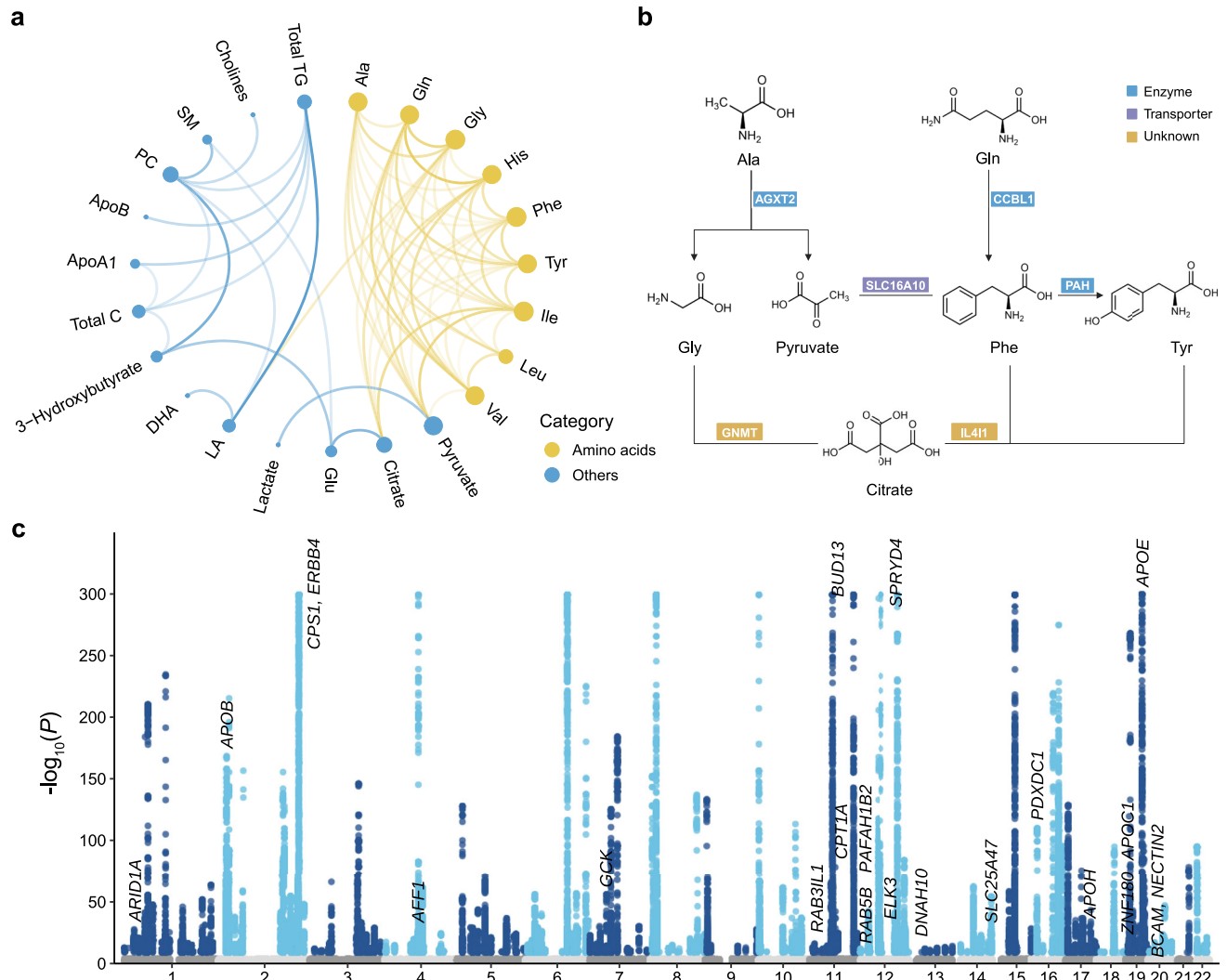

**Fig. 3 | Generation of 64 metabolite ratios and visualization of their GWAS results. a** The chord diagram depicts the generation of 64 biologically plausible metabolite ratios. Each connection represents a pathway of ratio derivation. Amino acids are denoted by yellow nodes, while metabolites in other categories are represented by blue nodes. The size of each node represents the number of metabolite ratios involving that metabolite. The darker the connecting lines, the higher the number of significant genome-wide associations for the metabolite ratios. Source data are available in Supplementary Data 5. **b** Examples of metabolite pairs sharing common proteins as documented in the HMDB database. Source data

are available in Supplementary Data 5. **c** Manhattan plot displaying the GWAS results for 64 metabolite ratios. Each point represents a trait-associated locus, with its corresponding $-\log_{10}(P)$ value on the y-axis and its chromosomal position on the x-axis. Statistical analysis was performed using an additive model of linear regression implemented in PLINK2.0. Significant variants (two-sided $P < 1.6 \times 10^{-10}$, derived from Bonferroni correction for 313 metabolic traits) are shown in colors, while insignificant ones are displayed in gray. The $-\log_{10}(P)$ values are capped at 300. See abbreviations in Supplementary Data 2 and 5.

fine-mapped causal variants, while no potential causal variants were detected for isoleucine to leucine ratio. The 3610 potential causal signals accounted for 0.03% (isoleucine to valine ratio) to 15.21% (omega-3 fatty acids) of the residual trait variance (Fig. 5b and Supplementary Data 11). Each causal variant explained 0.002% to 11.78% of the phenotypic variance (median = 0.09%), with 244 variants explaining ≥ 1% (Supplementary Fig. 5).

The putative causal variants identified through fine-mapping may exert their effects through multiple independent genetic mechanisms. For instance, at the *MLXIPL* locus, we revealed several alleles associated with genetic regulation of the triglycerides to total lipids in very small VLDL percentage. Through fine-mapping, we pinpointed three primary causal signals: an intronic variant in *MLXIPL* (rs13234378, $\beta = -0.16$, $P = 2.97 \times 10^{-228}$), and two intergenic variants near *TBL2* (rs573252567, $\beta = -0.15$, $P = 1.93 \times 10^{-174}$) and *VPS37D* (rs34958196, $\beta = -0.15$, $P = 3.44 \times 10^{-215}$) (Fig. 5c). Notably, among the likely causal variants identified in this locus, one was a lead variant pinpointed through clumping

analysis while the other two were not. Functional annotation using the FAVOR database[31] revealed that rs13234378 and rs573252567 are located in active enhancer flanking regions (EnhAF), while rs34958196 resides in a poised promoter region (PromP), suggesting potential regulatory roles for these noncoding variants.

To investigate how lipid-related loci affect lipoprotein metabolism, we focused on putative causal variants associated with unsaturation degree of fatty acids. This trait may reduce cardiovascular risk by altering lipoprotein composition, increasing HDL cholesterol and lowering low-density lipoprotein (LDL) cholesterol[32,33]. We standardized effect estimates to range from −1 to 1 by dividing each estimate by the strongest association effect estimate observed across all metabolic traits within each region, minimizing the influence of statistical strength and highlighting overall patterns[10]. Distinct patterns emerged between the variants and cholesterol levels (Fig. 5d, Supplementary Data 12). For instance, the missense SNP rs77960347, which is putatively damaging and leads to a loss of function of *LIPG*,

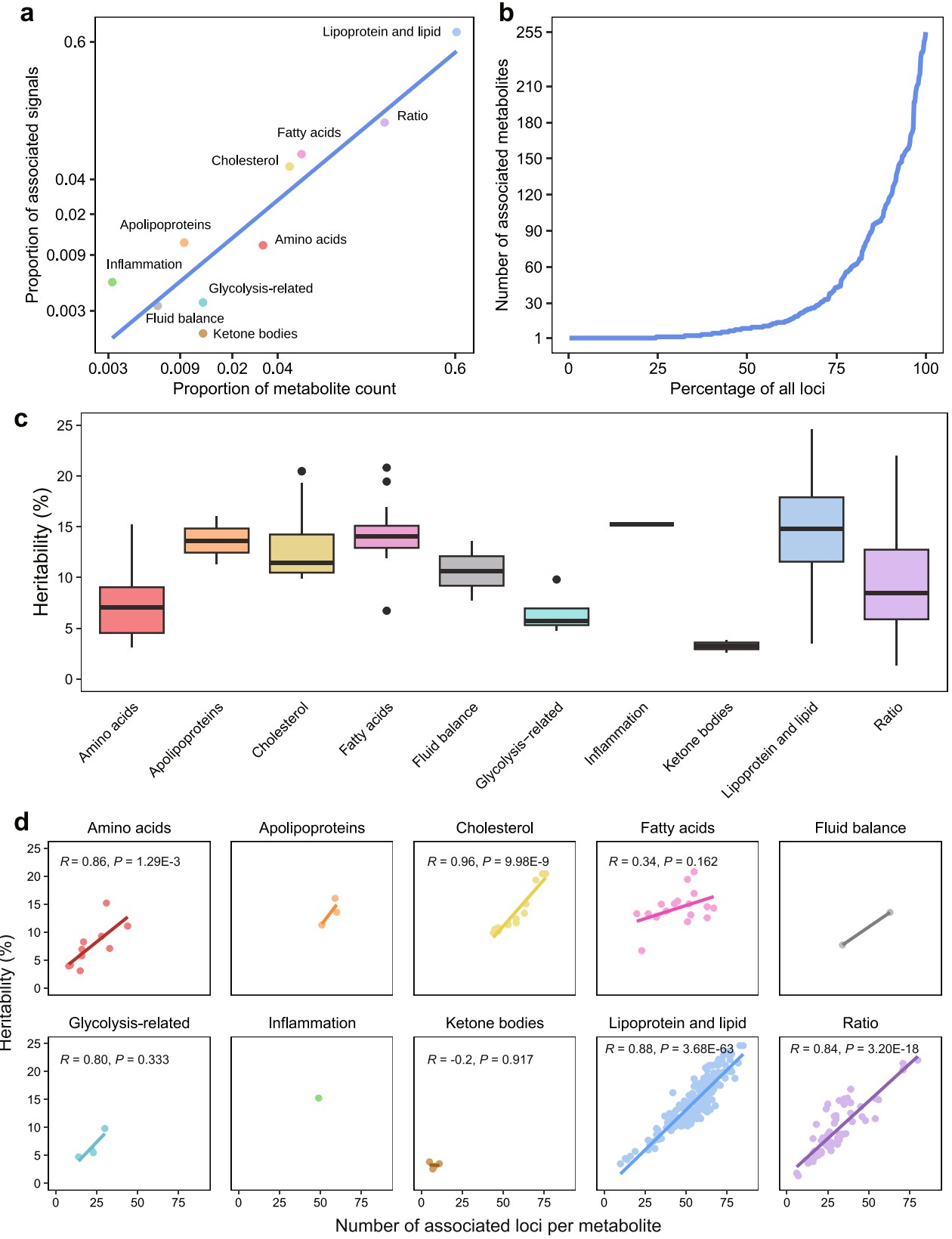

was positively associated with cholesterol levels across all lipoproteins. Endothelial lipase, encoded by *LIPG*, preferentially hydrolyzes HDL cholesterol over other lipoproteins[34,35], supporting our finding of a stronger association between rs77960347 and HDL cholesterol. On the other hand, rs174575, 11:61591995_GAA_G, and rs174564 in *FADS2* were associated with increased cholesterol levels in larger VLDL, but with

decreased cholesterol levels in smaller VLDL, LDL and larger HDL. These three intronic variants may exert regulatory effects, as they are located in promoter (marked by H3K4me3 and H3K9ac) and enhancer (marked by H3K4me1 and H3K27ac) regions across multiple tissues, according to the Roadmap Epigenomics project[36]. Fatty acid desaturase, encoded by *FADS2*, regulates fatty acids unsaturation[37] and has

**Fig. 4 | Genetic architecture of metabolites and derived ratios. a** Correlation between the number of metabolites and the number of significant genetic associations (Bonferroni correction for total metabolic traits, $5 \times 10^{-8}/313 = 1.6 \times 10^{-10}$) within ten major categories. Data are plotted on a log scale, with each point representing a unique category of metabolites. Source data are available in Supplementary Data 3. **b** Distribution of the number of associated metabolites per genetic locus, illustrating the pleiotropic effects on metabolite profiles. The x-axis represents the percentage of all loci, while the y-axis quantifies the number of metabolites associated with each locus. See details in Supplementary Data 4 and 8. **c** Box plots illustrating the heritability estimates for metabolic traits grouped by biochemical categories. Colors distinguish different metabolite categories. Boxes represent the IQR, with the horizontal line indicating the median. Whiskers extend to the most extreme data point that is no more than 1.5 × IQR from the edge of the box, with points beyond representing outliers. The numbers of metabolites for each category are provided in Supplementary Data 2. Source data are available in Supplementary Data 9. **d** Scatterplots for each biochemical category, illustrating the relationship between the heritability of metabolites and the number of associated genetic loci. The fitted lines were derived from a linear regression in each facet, with corresponding Spearman's correlation coefficients and two-sided *P* values provided, if applicable. See details in Supplementary Data 4 and 9.

been implicated in atherosclerotic cardiovascular diseases[38,39]. The lipid profile alterations linked to these variants may explain *FADS2*'s role in cardiovascular risk, though the underlying mechanisms for the differential effects across lipoprotein subclasses warrant further experimental investigation. Overall, genes containing these variants, such as *LIPG*, *FADS2*, *PCSK9*, and *ANGPTL4*, play key roles in cholesterol and fatty acid metabolism, glucose homeostasis, and insulin sensitivity[34,40–44]. These findings further advance our understanding of the complex effects of genetic variants on lipoprotein metabolism.

### GWAS validation and novelty assessment

To identify previously reported associations and assess whether they were replicated in our study, two methods were employed. First, we compiled a list of 19 representative metabolomics GWAS with overlapping metabolites and genomic variants to our study (Supplementary Data 13). For each of these studies, we compared our estimated effect sizes for each significant variant against those reported in earlier GWAS studies ($P < 5 \times 10^{-8}$ or more stringent). This comparison demonstrated substantial concordance across the datasets, with a median Spearman's correlation of 0.98 (IQR: 0.90–0.99), and 1164 of 1327 pairwise comparisons achieving Bonferroni-corrected significance ($P < 0.05/1327 = 3.77 \times 10^{-5}$; Supplementary Data 14, Supplementary Fig. 6). Correlations were higher for studies including UKB samples (median R = 0.99) than for those without (median R = 0.93). Sensitivity analyses using less stringent thresholds yielded consistent results (Supplementary Data 15). Of the 3610 potential causal associations identified, 2785 were previously reported, 40 were in LD with previously known variants, and 785 associations were considered novel.

Second, we compared our results with those from the latest published GWAS study on circulating NMR metabolic traits, which included 136,016 participants across 33 cohorts[10]. We observed that 73% of the lead variants identified in this study were validated in our results ($P < 5 \times 10^{-8}$ and consistent effect direction), whereas 47% of our lead variants were validated by this study (Supplementary Data 16).

We also performed a validation analysis on the lead variants identified in the discovery cohort using samples from 36,445 individuals of non-British ancestries. While the effect sizes were highly correlated with those from the discovery cohort (median Spearman's correlation coefficient R = 0.96, IQR 0.94–0.97, Supplementary Data 17), only 24% of the variants in the validation cohort reached genome-wide significance and had the same effect direction as in the discovery dataset, which may be partly attributed to the smaller sample size[10]. A pan-ancestry meta-analysis of the lead variants largely confirmed the primary results (median Spearman's R = 1.00), with 85% of variants showing more significant *P* values than in the primary analysis.

### Identification of potential effector genes

Metabolites are not directly encoded by genes, but are linked to gene-encoded proteins through various biological processes including synthesis, degradation, conversion, secretion, and transport. To identify the potential effector genes for metabolites, evidence should be integrated from multiple dimensions. For each putative causal variant, we first retrieved the 20 nearest protein-coding genes within a 1-Mb genomic region. These candidate genes were filtered based on biological evidence from multiple specialized databases, and co-localization analyses with eQTL and pQTL summary statistics (Methods). Genes with both biological and colocalization evidence were designated as effector genes, leading to the annotation of 71 variant-metabolite pairs. For the remaining 3539 pairs not annotated through this approach, the nearest genes were assigned.

In total, we assigned 221 unique genes to 312 metabolic traits (Supplementary Data 18 and 19). Notably, we linked *GCKR* to 205 metabolites across diverse categories based on genomic proximity, suggesting its well-established role in glucokinase regulation may have far-reaching metabolic implications beyond glucose homeostasis. We also identified novel genes closely associated with metabolites. For instance, *LDHB* encodes the B subunit of lactate dehydrogenase, which catalyzes the reversible conversion of pyruvate and lactate[45]. *LDHB* eQTLs colocalized with both pyruvate levels and the lactate to pyruvate ratio. The variant rs138560021 emerged as the likely shared causal variant, showing negative associations with *LDHB* expression ($\beta = -0.33$, $P = 9.82 \times 10^{-7}$) and the lactate to pyruvate ratio ($\beta = -0.10$, $P = 1.22 \times 10^{-13}$), and positive association with pyruvate levels ($\beta = 0.09$, $P = 2.05 \times 10^{-13}$). These consistent directional effects suggest that reduced *LDHB* expression may disrupt pyruvate-lactate balance, in line with the enzyme's known biological function.

To understand the clinical and pharmacological relevance of the 221 effector genes, we systematically searched these genes across reference databases. We identified 89 associated Mendelian diseases from the Online Mendelian Inheritance in Man (OMIM) database[46], 471 drugs at different development phases from the DrugBank database[47], and 124 phenotypic changes in knockout mice from the International Mouse Phenotyping Consortium (IMPC) database[48] (Supplementary Data 20).

### Rare protein-coding variants associated with metabolic traits

Next, we aimed to identify rare coding variants associated with metabolic traits, as they often confer larger effect sizes and are more likely to implicate causal genes. We leveraged WES data from 197,774 UKB participants of white British ancestry, yielding 10,662,863 rare variants after strict quality control.

We conducted rare-variant aggregation tests using ten models based on two functional categories (Loss-of-function [LOF] variants alone or in combination with missense variants), five MAF bins (1%, 0.1%, 0.01%, 0.001%, and 0.0005%), as well as ten gene-based tests including burden, ACAT, SKAT, and their respective omnibus or joint tests (Methods). Overall, we identified 2948 significant gene-metabolite pairs, involving 126 genes across 308 metabolic traits, at a Bonferroni-corrected threshold of $0.05/(313 \times 19,559) = 8.17 \times 10^{-9}$ (Supplementary Fig. 7, Supplementary Data 21). Of these, 1,804 signals were identified using the LOF mask, 2827 using the combined LOF and missense variant mask, and 33 were uniquely identified through joint tests in REGENIE, which integrate burden masks generated from these two annotation classes as well as all MAF thresholds. The higher carrier

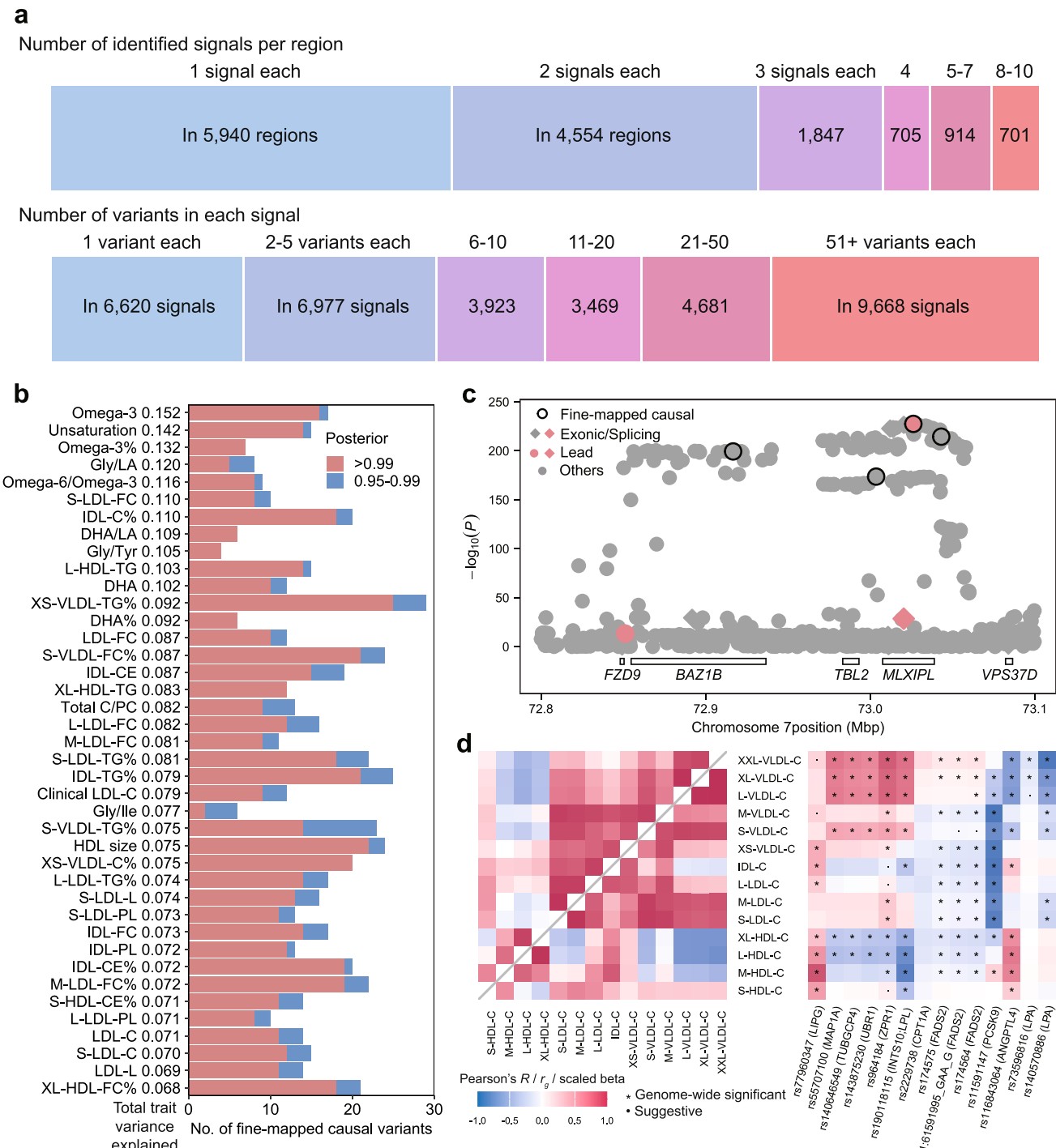

**Fig. 5 | Summary of fine-mapped associations across 313 metabolic traits.**
**a** FINEMAP summary. At the top, the numbers of association signals identified in each region that contains at least one independent association are shown. This includes a single signal in 5940 regions and 2–10 signals in 8721 regions across 313 metabolites. At the bottom, the numbers of candidate causal variants in the credible set with ≥ 99% posterior probability are displayed. For example, 6620 signals were mapped to a single variant in the credible set across 313 traits. Source data are available in Supplementary Data 10. **b** Breakdown of the number of fine-mapped variants. Red bars represent causal variants with a posterior probability greater than 0.99. Blue bars represent causal variants with a posterior probability between 0.95 and 0.99. The x-axis shows the number of fine-mapped causal variants, and the y-axis indicates the total variance explained by those variants with a posterior probability greater than 0.99. The 40 traits with the highest total variance explained are displayed. Source data are available in Supplementary Data 11. **c** A regional association plot showing the fine-mapping result for XS-VLDL-TG% at the

*MLXIPL* loci, based on GWAS summary statistics. Fine-mapped variants are indicated by black outlined points. Exonic or splicing functional variants are represented as diamonds. Lead variants are highlighted in red. The raw two-sided *P* values are provided. **d** Heatmaps of phenotypic (top left inset) and genetic (bottom right inset) correlation structure of cholesterol in lipoprotein subclass particles (left) and the association landscapes of variants associated with degree of unsaturation (right). Pairwise Pearson's correlation coefficients and genetic correlations (left), and effect estimates for the variant–metabolic trait associations (right) are represented as a color range. Stars indicate genome-wide significance and dots indicate suggestive associations. The variant effect sizes were scaled relative to the absolute maximum effect size in each region. The effect allele for each variant is shown. Each column represents a variant, and each row corresponds to a metabolic measure. Two-sided tests were used for statistical analysis. Source data are available in Supplementary Data 12. See abbreviations in Supplementary Data 2 and 5.

frequency of missense variants in the population compared to LOF variants likely contributes to the increased discovery power when both are combined. Among all aggregation tests, burden tests identified the most associations (n = 2712), including 171 unique to this method. ADD-SKATO-ACAT ranked second with 1169 associations, 104 of which were unique. The strong performance of the burden test suggests that rare coding variants associated with metabolites generally have consistent effect directions and magnitudes, aligning with the assumptions of this test. A total of 1910 gene-trait associations were replicated in both male and female subgroups (P < 0.05/2948). The numbers of genes associated with each trait ranged from 1 to 20, with a median value of 10. Exome-wide aggregate-based analyses showed good calibration of P-values across all performed tests, as indicated by genomic inflation factors consistently below 1.06 (Supplementary Data 22). We then compared our findings to published metabolite GWAS, and the lead variant-trait associations identified in our GWAS. Notably, examining the genes annotated for single variant associations, we showed that of the 2948 gene-trait pairs identified, 1778 had associations involving common variants in previous GWAS, 1408 had associations involving lead variant identified in our GWAS, and the remaining 939 pairs provided new insights that complemented the GWAS findings. These results underscore the added value of WES, potentially benefiting from its deep coverage and enhanced sequencing accuracy in protein-coding genes.

We then performed conditional analyses to determine whether our signals were independent of nearby common variants. We modeled our gene-trait pairs while conditioning on nearby (< 1 Mb) LD-independent common variants identified in our GWAS. Among the 963 gene-trait pairs in which at least one nearby common associated variant (±1 Mb) was directly mapped to the WES genotype, the effect sizes and P values were not substantially attenuated in 904 pairs (93.87%, Supplementary Data 21), indicating the independent role of rare variants on metabolic traits.

We compared our findings with a recent WES study of metabolic traits involving 99,283 UKB participants by Nag et al.[19]. Among the 244 metabolites analyzed in both studies, we identified 2524 significant gene–metabolite associations. Of these, 1357 (53.76%) overlapped with those reported by Nag et al., while 1167 (46.24%) were novel (Supplementary Data 23). Conversely, of the 1476 associations reported by Nag et al., 1357 (91.94%) were confirmed in our analysis. These results demonstrate both the robustness of our approach in replicating known associations and its power to uncover novel gene–metabolite pairs.

### Genetic insights into metabolite-disease associations

To explore the disease relevance of the plasma metabolome, we examined genetic correlations between 313 metabolic traits and 2179 clinical traits (Supplementary Data 24), uncovering 75,927 metabolite-disease pairs with genetic overlap (Supplementary Fig. 8). The extensive genetic correlations prompted further colocalization analysis, resulting in the identification of 72,538 metabolite-disease pairs with shared genetic determinants (Supplementary Data 25). Next, we conducted two-sample MR analyses for the colocalized pairs. Notably, reverse MR analysis revealed bidirectional causal relationships for half of the significant main MR results, including reverse causation between type 2 diabetes (T2D) and glucose. We ultimately identified 36 potentially causal associations that remained significant in both main and sensitivity MR analyses while showing no evidence of reverse causality (Methods, Fig. 6a, Supplementary Data 26–29). Among those, glucose associated with the most diseases (n = 9), followed by acetate (n = 4), the glucose to sphingomyelins ratio (n = 3) and free cholesterol in large HDL (n = 3).

We replicated well-known relationships, such as the positive associations between glucose and gestational diabetes, supporting the robustness of our analytical framework. Similarly, we found that free cholesterol in large HDL was negatively associated with cholelithiasis, in line with previous observational studies reporting that dyslipidemia often accompanies gallstone formation[49]. Beyond these known associations, we also identified several less-studied findings. Notably, strong evidence of colocalization was observed between acetate and atrial fibrillation and flutter at rs4766897 in ACAD10. MR analysis demonstrated that genetically predicted plasma acetate levels were associated with a reduced risk of atrial fibrillation and flutter (odds ratio [OR] = 0.60, P = 2.07 × 10^{-8}), suggesting a potential beneficial role of acetate in cardiovascular diseases, which was also reported in a previous study[50] (Fig. 6b, c).

Beyond individual metabolites, our analysis extended to explore relationships between metabolite ratios and various diseases. Notably, genetically predicted values of the isoleucine to leucine ratio were positively correlated with height (OR = 1.28, P = 2.10 × 10^{-10}), with strong evidence of colocalization at rs1260326 in GCKR. Isoleucine and leucine, two branched-chain amino acids (BCAAs), play crucial roles in skeletal muscle metabolism and growth regulation[51,52]. However, the imbalance of BCAAs may adversely affect growth. This phenomenon aligns with a previous study showing that excessive leucine levels, which lead to relatively isoleucine deficiency, impaired the growth of chicks[53]. Furthermore, we found that higher genetically predicted levels of cholesterol to total lipids percentage and cholesteryl esters to total lipids percentage in large HDL were associated with a reduced risk of ischemic heart disease. Additionally, cholesterol to total lipids percentage in large HDL was inversely associated with the risk of coronary atherosclerosis.

## Discussion

This study presents a large-scale GWAS and WES investigation of NMR-based metabolomics, analyzing 249 metabolic measures and 64 biologically plausible ratios in over 250,000 participants. A total of 24,438 independent variant-metabolite associations across 427 genomic loci were identified through GWAS, with effect sizes highly concordant with 19 previous studies. The fine-mapping analysis further prioritized 3610 potential causal associations (785 novel), providing a refined set of genetic markers likely to have a direct impact on metabolic traits. In addition, 45 potential effector genes were assigned to 71 variant-metabolite pairs via integration of biochemical and colocalization evidence, providing biological context for the identified variants to facilitate further mechanistic studies. Integrating colocalization and bidirectional MR analyses, we identified 36 potential causal metabolite-disease associations after rigorously excluding pleiotropic loci and ruling out reverse causality. Furthermore, rare variant effects were characterized through WES-based aggregate testing, with the majority found to be independent of nearby common variants. Taken together, this study unveils the intricate genetic architecture underlying plasma metabolic traits encompassing both rare and common variants and provides valuable leads regarding the potential roles of metabolism in disease pathogenesis.

Prior studies examining metabolite ratios were limited by smaller sample sizes, different metabolic profiles, or hypothesis-free approaches that complicated result interpretation[11-16]. Although several recent preprints have conducted metabolomics studies using UKB data, they have not explored the genetic architecture of metabolite ratios, leaving a critical gap that our study addresses[54-56]. Our study builds on existing research by incorporating 64 biologically plausible ratios into a large-scale analysis of over 250,000 individuals. The GWAS analysis of these ratios revealed genetic variants undetectable with individual metabolites. For instance, we constructed the leucine to valine ratio based on their shared regulation by the branched-chain amino acid transferases BCAT1 and BCAT2, which catalyze their transamination into corresponding keto acids[57]. The GWAS analysis of this ratio identified four independent lead variants (rs11047639, rs7965671, rs149598519, rs5797103) near BCAT1 or BCAT2. While these

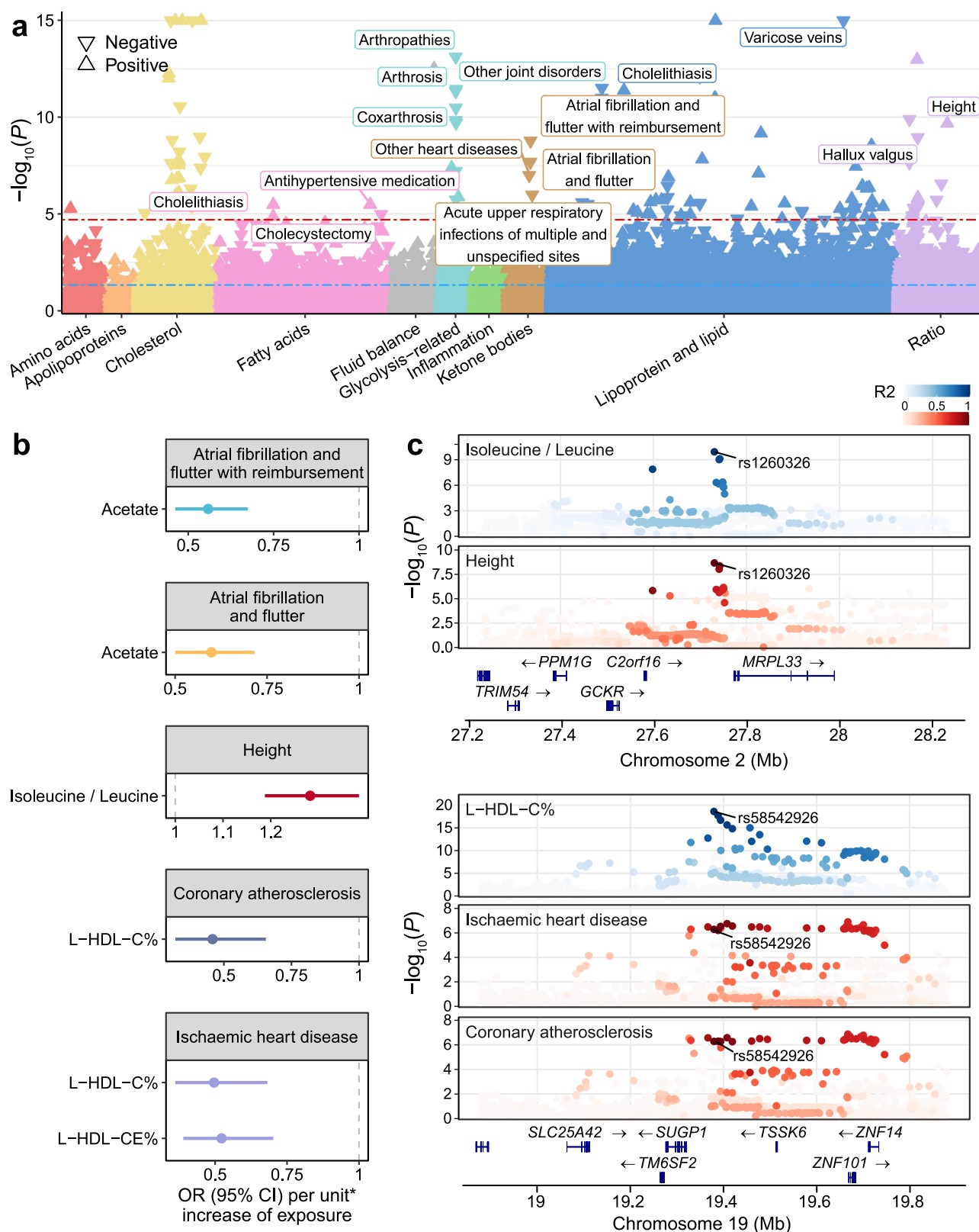

variants showed no significant associations with leucine or valine, they demonstrated significance in the ratio analysis ($P < 1.26 \times 10^{-11}$, $P$-gain > $2.62 \times 10^{8}$). This likely reflects the ratio's ability to normalize confounding factors or amplify subtle substrate preference effects, contributing to enhanced statistical detection power.

The intricate metabolic network typically involves multiple interacting genes, making gene annotation based solely on genomic position insufficient to capture the underlying complexity. By integrating QTL colocalization and biochemical evidence, we assigned 45 unique genes to 71 potential causal associations. When combined with medical and pharmacological knowledge, these genes may help uncover mechanistic links between metabolites and diseases. Notably, *GLUD1* was assigned to 12 amino acid-related associations (9 novel), including those involving glutamine and its related ratios. Among

**Fig. 6 | Genetic insights into metabolite-disease associations. a** Manhattan plot showing the association results from main MR analyses of metabolite-disease pairs with colocalization evidence. Metabolites are labeled on the x-axis, with color coding to distinguish different categories. Statistical analysis was performed using the Wald ratio method for single-instrument cases and the IVW method when multiple instruments were available. For each metabolite category, the four most significant diseases that remained robust in sensitivity analyses and were not significant in reverse MR are labeled. The $-\log_{10}(P)$ values are presented on the y-axis, which is capped at 15. Significance thresholds are marked by a blue dashed line at $P < 0.05$ and a red dashed line indicating corrections for multiple testing (FDR–corrected $Q < 0.05$, equivalent to $P < 2.09 \times 10^{-5}$). Arrows reflect the impact on disease risk per unit increase in metabolic trait, with upward arrows indicating increased risk and downward arrows suggesting decreased risk. Source data are available in Supplementary Data 26. **b** Forest plots depicting results for metabolite-disease pairs of interest. Odds ratio estimates (central points), 95% confidence intervals (error bars) and two-sided $P$ values were derived from two-sample MR analysis using the IVW method. An asterisk (*) indicates one standard deviation of the log-normalized value of the metabolic traits. Source data are provided in Supplementary Data 26. **c** Stacked regional plots illustrating the colocalization (PP.H4 > 0.8) of metabolite-disease pairs. Colocalization analysis was performed using coloc R package with Bayesian approach. A two-color gradient displays LD ($R^2$) with the candidate causal variant pinpointed through colocalization analysis. Blue represents metabolic traits, while red indicates clinical diseases. $P$-values shown are two-sided and not adjusted for multiple comparisons. Source data are provided in Supplementary Data 25. See abbreviations in Supplementary Data 2 and 5.

these, 4 annotations were supported by biological and colocalization evidence, while the remaining were assigned based on genomic proximity. *GLUD1* encodes glutamate dehydrogenase, which promotes the metabolism of glutamate and glutamine to generate ATP, stimulating insulin secretion[58]. Activating mutations in *GLUD1* can lead to hyperinsulinism-hyperammonemia syndrome (OMIM 606762). The identified genetic associations further support *GLUD1*'s central role in regulating amino acid levels, potentially linking its enzymatic activity to insulin dysregulation in pathological contexts.

Our study also leveraged WES data to investigate independent effects of rare variants on metabolic traits. Through aggregate testing, we identified 2948 gene-trait pairs. Of these, 1408 had associations involving lead variants identified in our GWAS, thereby reinforcing the GWAS findings. WES additionally provided novel insights absent in GWAS. For instance, although *HMGCS2* encodes the enzyme catalyzing the first step of ketogenesis[59,60], our GWAS did not identify significant associations between variants in *HMGCS2* and acetoacetate. In contrast, rare missense variants in *HMGCS2* were significantly associated with acetoacetate in WES. Similarly, *BCKDK*, which encodes a kinase regulating the mitochondrial BCKDH complex involved in BCAA catabolism, was linked to isoleucine and total BCAA levels in WES but not in GWAS. Additionally, our study demonstrated that most WES-based findings were conditionally independent of nearby common variants. This combined approach offered a more comprehensive understanding of the genetic regulation of metabolic traits.

Dysregulation of metabolism is a prevalent characteristic across a diverse spectrum of human diseases[1,2,61]. By combining colocalization and MR findings, this study highlighted 36 putative causal metabolite-disease associations. Notably, we identified a novel protective association between elevated acetate levels and reduced atrial fibrillation and flutter risk. This observation aligns with previous experimental findings demonstrating acetate-mediated cardioprotective effects, including attenuation of hypertension, cardiac hypertrophy, and cardiorenal fibrosis in animal models[62]. Our human genetic evidence strengthens the translational potential of these observations, suggesting potential therapeutic applications of acetate supplementation in cardiovascular disease prevention. Meanwhile, the bidirectional MR analysis revealed that disease onset may lead to metabolic disturbance, providing new mechanistic insights into disease pathogenesis. For example, the bidirectional relationship between glucose and T2D suggests that maintaining glucose homeostasis may represent a key therapeutic target, while elevated glucose levels could serve as a biomarker for disease progression. Collectively, these findings highlight the potential clinical value of the plasma metabolome, underscoring the importance of metabolic regulation in human health and disease.

Our study still has several limitations. First, using non-fasting plasma samples may have introduced variability, although we have adjusted the metabolites for the duration between the participants' last food or drink intake and blood sample collection. Second, NMR-based metabolomics captures fewer metabolites compared to MS,

with a predominant focus on lipid-related traits. This may partly explain why most identified metabolite-disease associations are enriched in metabolic and cardiovascular categories and limits valid comparisons of heritability across metabolite classes. Nevertheless, the cost-effectiveness of NMR enabled the UKB cohort to include over ten times more participants than the largest MS-based cohort[8]. Furthermore, although our gene annotation workflow integrated multiple evidence sources, most genes were annotated based on genomic proximity due to current database limitations. This approach also assumed that genetic variants influence metabolites through differential regulation of gene expression and protein abundance, potentially overlooking genes where variants directly affect protein structure, stability, or catalytic activity. Finally, although we have excluded instruments with obvious pleiotropic effects, the influence of potential horizontal pleiotropy on MR findings might not be entirely ruled out.

In summary, our study's substantial increase in sample size and comprehensive metabolite profiling yielded 785 novel putative causal signals and 45 potential effector genes with biological and colocalization evidence. By comparing our GWAS results in the discovery analysis with those from non-British participants and 19 previous studies, we further confirmed the reliability of our findings. Through systematic annotation and integration with disease genetics, we identified metabolite-disease associations with shared causal variants. Further integration with WES data uncovered 2948 gene-trait pairs, the majority of which were independent of nearby common variants.

We believe this work greatly advances our understanding of the genetic basis underlying the plasma metabolome, while our companion phenotypic investigation demonstrates the clinical utility of metabolomics for disease diagnosis, prediction and treatment target identification[63]. Together, these complementary efforts provide an integrated resource linking genetic regulation, metabolic variation, and human disease, facilitating future mechanistic discoveries and translational applications.

## Methods
### Study cohort

The study design and conduct complied with all relevant regulations regarding the use of human study participants and was conducted in accordance with the criteria set by the Declaration of Helsinki. Ethics approval was granted by the North West Multi-Centre Research Ethics Committee. All participants provided informed consent via electronic signatures. Our research was carried out under UKB application number 19,542.

UKB is a population-based prospective cohort consisting of more than 500,000 individuals aged 40 to 69 years at baseline[23]. Participants were enlisted from 22 assessment centers across the UK between 2006 and 2010 and underwent extended monitoring. In this study, we excluded participants who were officially withdrawn by UKB or had more than 20% missing metabolite values. This resulted in a final sample of 254,825 individuals with both metabolomic and genomic

data available (Supplementary Data 1). Among these, 218,380 were of white British ancestry (comprising 189,846 with imputed genotype data and 197,774 with WES data), and 36,445 were of non-British ancestry with imputed genotype data.

## Metabolites and metabolite ratios processing

**Metabolomic profiling.** The NMR-based metabolomic profiling of 275,000 UKB participants at baseline was utilized in our study (UKB Category 220). Details of the technical methodology are available at (https://biobank.ndph.ox.ac.uk/showcase/ukb/docs/NMR_companion_phase2.pdf). Briefly, a total of 249 metabolic measures were quantified simultaneously for each plasma sample. The metabolic profile includes routine cholesterol, fatty acids, lipoprotein subclass profiling with lipid concentrations across 14 subclasses, and low-molecular-weight metabolites, such as ketone bodies, amino acids, and glycolysis-related metabolites (Supplementary Data 2). To augment the breadth of our investigation, we additionally defined 64 metabolite pairs based on their shared biochemical activities, specifically the enzymes, transporters or other common proteins involved, as documented in the HMDB database (Supplementary Data 5). The 249 primary and 64 derived metabolic traits were then systematically subjected to quality control and genetic analysis as described below.

**Statin identification and adjustment.** Given that plasma metabolites are potentially affected by drug administration, especially statins, we conducted an examination of pharmaceutical interventions with a specific focus on statin usage (Field ID: 20003). For participants who initiated statin therapy between their initial enrollment (2006–2010) and their first repeat assessment (2012–2013), we assessed the impact of statin use on metabolite levels. Specifically, for each metabolic trait, we performed standard linear regression to test whether the difference between pre- and post-statin measurements was significant, adjusting for sex, baseline age, age difference between baseline and follow-up, socio-economic status (as measured by the Townsend Deprivation Index, TDI), body mass index (BMI), BMI difference between baseline and follow-up, the first 20 principal components (PCs) of genetic ancestry, and interactions between age, BMI, and sex, using the following formula:

$$y_{\text{post statin}} = \beta_0 + \beta_1 y_{\text{pre statin}} + \beta_2 \text{sex} + \beta_3 \text{age}$$
$$+ \beta_4 \Delta\text{age} + \beta_5 \text{TDI} + \beta_6 \text{BMI} + \beta_7 \Delta\text{BMI}$$
$$+ \beta_8 \text{PC}_1 + \beta_9 \text{PC}_2 + \cdots + \beta_{27} \text{PC}_{20} + \beta_{28}(\text{age} \times \text{sex}) \quad (1)$$
$$+ \beta_{29}(\text{BMI} \times \text{sex}) + \beta_{30}(\text{age} \times \text{BMI}) + \epsilon$$

Only metabolites demonstrating a significant change associated with statin use ($P < 0.05$) were considered for subsequent correction. For each of these traits, we calculated the ratio of post-statin to pre-statin metabolite levels across participants and derived the mean ratio to serve as a correction factor. Finally, this correction factor was applied to adjust the metabolite measurements of participants who were on statin therapy upon baseline assessment[4], using the following formulas:

$$C = \frac{1}{n}\sum_{i=1}^{n}\frac{y_{\text{Post statin}}, \text{i}}{y_{\text{pre statin}}, \text{i}} \quad (2)$$

$$y_{\text{adjusted}} = \frac{y_{\text{baseline on statin}}}{C} \quad (3)$$

## Imputed genome-wide genotypic data in the UKB

Imputed genome-wide genotypic data were accessible for 488,377 UKB participants. Genotyping was performed using two interconnected Affymetrix arrays: the UK BiLEVE Axiom Array for 9.9% of participants and the UKB Axiom Array for the other 90%. The Haplotype Reference Consortium (HRC) data and UK10K haplotype

resources were utilized as imputation reference panels[64]. The procedures for DNA collection, sample processing, and quality control have been detailed elsewhere[64]. In addition to sample quality control by the UKB team, we further excluded individuals due to sex mismatches, missing data rates exceeding 5%, unusual heterozygosity rates, and the presence of more than ten probable third-degree relatives. Additionally, variants with call rates under 95%, MAF below 1%, or significant Hardy-Weinberg deviations ($P < 1 \times 10^{-6}$) were excluded, retaining 7,924,871 variants for the downstream analysis.

## Genome-wide association analysis

Metabolite values that deviated by more than four times the interquartile range from the median were removed. Additionally, a linear regression was established, wherein the $\log_{10}$-transformed metabolites served as the dependent variables, while the covariates acted as the independent variables. Specifically, the covariates included age, ethnicity, sex, fasting time, month of assessment, genotype measurement batch, the top 40 genotype PCs, age indicators by sex interactions, and ethnicity by sex interactions. Sex was determined based on self-reported data via questionnaires. The scaled residuals derived from this model were then employed as the refined predictors in the following genetic association studies. The data after quality control was highly correlated with the original data for each metabolite analyzed (Supplementary Fig. 9 and Supplementary Fig. 10).

For each metabolic trait under investigation, GWAS analysis was conducted on the previously computed scaled residuals. UKB imputed data were converted from BGEN to BED format using PLINK2.0, generating best-guess genotypes by selecting the most probable genotype per variant. The GWAS analysis employed an additive model of linear regression implemented in PLINK2.0[65]. All the $P$ values from the association analysis were from two-sided tests. To account for multiple comparisons, the significance threshold was defined as the genome-wide threshold divided by the number of metabolic traits assessed ($5 \times 10^{-8}/313 = 1.60 \times 10^{-10}$). We used Fuji plot to visualize the single-locus level pleiotropy and the polygenicity of metabolic traits[66,67].

## Clumping analysis and definition of genomic locus

To identify independent genetic variants, we clumped the white British GWAS results using PLINK2.0[65], informed by LD information available from a random sample of 5000 unrelated UKB participants included in the GWAS analysis. The following parameters were used: --clump-p1 1.6e-10 --clump-p2 1.6e-10 --clump-r2 0.01 --clump-kb 10,000. Next, to assess pleiotropy at the single-locus level, we defined genomic loci as regions extending ±250 kb around each lead variant. In the subsequent step, we merged overlapping loci for all metabolic traits.

## Evaluation of denominator-driven effects and unique loci in ratio associations

We evaluated whether ratio associations were statistically stronger than those for individual metabolites using the $P$-gain metric (defined as $\frac{\min(P_{\text{denominator}}, P_{\text{numerator}})}{P_{\text{ratio}}}$, significance threshold set at $64/(2 \times 0.05) = 640)$[13,68]. Loci uniquely identified through ratio-based analyses were determined using a Bonferroni-corrected threshold of $5 \times 10^{-8}/313 = 1.60 \times 10^{-10}$.

## LD score regression

We performed LD score regression using ldsc (version 1.0.1)[25] to assess the heritability of 313 metabolic traits. Additionally, we examined the genetic correlations between each pair of these metabolic traits, and between these traits and the 2179 diseases and traits included in the colocalization analysis. The default LD scores from the 489 unrelated European individuals in 1000 Genomes were utilized as our reference panel[69]. We converted our white British genetic statistics into LDSC format using munge_sumstats, aligning them with 1000 Genomes

Phase 1 variants that have ancestral allele calls in 1000 Genomes Phase 3.

To compute the LD score regression intercept and heritability, we executed ldsc.py with the following parameters: ldsc.py --h2 <trait summary statistics > --ref-ld-chr <ldsc/1000 G.EUR.QC/ > --w-ld-chr <ldsc/weights_hm3_no_hla/weights > .

Subsequently, to estimate the genetic correlation effects (significance threshold set at $P < 0.05$), we employed the genetic correlation mode with the following parameters: ldsc.py --rg <traits > --ref-ld-chr ldsc/1000 G.EUR.QC / --w-ld-chr ldsc/weights_hm3_no_hla/weights.

### Fine-mapping analysis

For each lead variant, we constructed a genomic region extending 1-Mb on either side. We merged overlapping regions associated with the same metabolite, resulting in 14,661 regions of 1.3 to 6.5 Mb in length. To pinpoint putative causal variants within these regions, we employed the FINEMAP software (version 1.4.21)[28], which leverages GWAS summary statistics and LD information to deduce putative causal variants through a Bayesian variable selection approach. The output generated by FINEMAP includes: (1) potential causal configurations with their corresponding posterior probabilities and Bayes factors, (2) the posterior probability and Bayes factor for every variant, and (3) credible sets for every detected causal signal. We employed FINEMAP using its standard settings, restricting the number of causal variants to a maximum of ten.

### Validation of GWAS results

To validate our GWAS findings against previous studies, we employed two approaches. First, we systematically searched PubMed, the NHGRI-EBI GWAS Catalog, and previous literature to compile a list of 19 metabolomics GWAS studies featuring metabolites that overlap with ours (Supplementary Data 13). Summary statistics were standardized to the hg19 reference genome, and metabolite names and alleles were harmonized. We compared effect sizes of significant variants ($P < 1.60 \times 10^{-10}$ in our study) with those reported in prior studies ($P < 5 \times 10^{-8}$ or stricter) using Spearman's rank correlation and linear regression when at least three variants overlapped. For each metabolite and study, we reported the Spearman's rank correlation coefficient, and the beta coefficient, standard error, and $P$ value from linear regression analysis. To assess robustness, we conducted sensitivity analyses using a range of significance thresholds of variants ($1.00 \times 10^{-5}$ to $1.60 \times 10^{-10}$). Second, we compared our results with the latest metabolomics GWAS study that measured metabolites in European ancestry cohorts[10]. We reported the proportion of lead variants replicated between this latest GWAS and our study, defining validation as having a $P$ value less than $5 \times 10^{-8}$ and matching the effect direction of the original study.

We also conducted a validation analysis of the lead variants identified in the British discovery cohort using individuals of non-British ancestries. Due to the limited sample sizes within each ancestry group, we combined the non-British samples for the validation. We assessed the consistency between the discovery and validation datasets using Spearman's rank correlation and reported the proportion of variants that remained genome-wide significant with consistent effect directions in the validation analysis.

### Novelty assessment of GWAS results

To ascertain whether the identified causal associations were previously documented for the same metabolic trait, we employed two approaches. First, we obtained the significant associations reported in 19 metabolomics GWAS publications (Supplementary Data 13). Second, we systematically curated published results from the NHGRI-EBI GWAS catalog database[70] (associations downloaded on 11 March 2024). This process led to the identification of 2785 known variant-metabolic trait associations that met a significance threshold of $P < 5.0 \times 10^{-8}$ or more

stringent. Additionally, we found 40 putative causal variants in LD ($r^2 > 0.8$)[16] with previously reported variants, which we categorized as potentially novel. The remaining 785 variant-metabolic trait associations were finally recognized as novel.

### Potential effector gene annotation

Using the human GENCODE resource[71], we initially retrieved protein-coding genes within a 1-Mb genomic region surrounding each of the unique causal variants. Then we further determined the nearest 20 protein-coding genes for each variant, based on the minimum distance from the variant to the genes' transcription start or end sites. These genes were referred to as candidate genes and subsequently underwent two steps of filtering.

First, we assessed the biological relevance of these candidate genes to the corresponding metabolites by querying three databases: the HMDB database[72], the Kyoto Encyclopedia of Genes and Genomes (KEGG) Pathway database[73] and the PubChem Chemical Co-occurrences in Literature database[74].

Second, we conducted co-localization analysis between GWAS signals of metabolites and eQTLs and pQTLs of candidate genes using coloc R package (5.2.3). We identified cis-eQTLs across 49 tissues (Supplementary Data 30) using gene expression data from the V8 release of the Genotype-Tissue Expression (GTEx) Project[75], and pQTLs were obtained from a previous study[76]. For each lead variant, GWAS and QTL summary statistics within 1-Mb were extracted as inputs. Prior probabilities were set as recommended[77]: $P_1 = 1 \times 10^{-4}$, $P_2 = 1 \times 10^{-4}$, and $P_{12} = 1 \times 10^{-5}$. PP.H4 $\geq 0.6$ was considered as evidence for colocalization[78].

Candidate genes listed in any of the queried databases as being associated with the metabolite were considered biologically relevant and were categorized as expression-relevant if they showed evidence of colocalization. Potential effector genes were subsequently defined as candidate genes that satisfied both biological relevance and expression relevance criteria. Multiple effector genes could be selected. If evidence was insufficient for all candidate genes due to limitations in current databases and QTL data, then the nearest gene was designated.

### Medical and pharmacological annotation

To gain insights into the medical and pharmacological implications of the identified potential effector genes, we performed comprehensive annotations utilizing multiple databases. Protein type information was retrieved from the UniProt database[79]. Mendelian traits and diseases were identified using the OMIM database[46]. Phenotypic changes observed in knockout mice were retrieved from the IMPC database[80], leveraging the available information on mouse strains, genetic knockouts, and associated phenotypic alterations. We also explored the DrugBank database[47] to identify drugs targeting the potential effector genes, providing insights into potential therapeutic interventions.

### Whole exome sequencing, quality control, and rare variant collapsing analysis

WES data were processed following the Regeneron Genetics Center (RGC) SPB pipeline. The sequencing, variant calling, and quality control procedures were previously described in detail[81]. We utilized the OQFE WES pVCF files aligned to the GRCh38 human reference genome build[82] and conducted further quality control as described in a previous study[83]. After splitting multi-allelic sites into bi-allelic sites, low quality and extreme outlier genotypes were removed using Hail. Variants with MAF $\geq 1\%$, call rate $< 90\%$, Hardy–Weinberg $P < 1 \times 10^{-15}$, and low-complexity regions in Ensembl were further excluded. Rare variants (MAF $< 1\%$) were annotated using SnpEff v4.347[84] against Ensembl Build 38.92, with the most severe consequence for each variant chosen across all protein coding transcripts. LOF variants were those

predicted to cause frameshift insertion or deletion, splice-site alteration, stop gain, and stop loss. Likely deleterious missense variants were defined as those predicted consistently to be deleterious by five in silico prediction tools including SIFT[85], Polyphen2_HVAR[86], LRT[87], Polyphen2_HDIV and MutationTaster[88]. Variants were collapsed for each gene for the gene-based collapsing tests. Participants who withdrew from the study, were duplicated, had discordance between self-reported and genetically inferred sex, had 2nd degree or closer relatives based on the KING-robust algorithm (kinship coefficient < 0.0884), or had Ti/Tv, Het/Hom, SNV/indel, and the number of singletons exceeding 8 standard deviations from the mean were removed. Finally, we restricted our main analysis to white British and excluded individuals with missing data rates exceeding 20%, resulting in a total of 197,774 participants for rare variant analysis.

Metabolite data were processed to exclude values that deviated by more than four times the interquartile range from the median. Exome-wide gene-based collapsing analyses were then performed using Regenie (version 3.4), adjusting for age, ethnicity, sex, fasting time, month of assessment, assessment center, the top 10 genotype PCs. Rank inverse normal transformation (RINT) was applied using the '--apply-rint' command. ACAT-O tests, SKAT-O tests, and burden tests were used, with age, ethnicity, sex, fasting time, month of assessment, assessment center, the top 10 genotype PCs being adjusted. To account for potential heterogeneity in the proportion and direction of causal variant effects, we employed ten complementary gene-based association tests to ensure robust detection of signals[89]: (1) ADD (Burden test): A conventional burden test that assumes all variants in the annotation set affect the phenotype in the same direction and with similar magnitude. (2) ADD-SKAT: The Sequence Kernel Association Test, which allows for heterogeneity in the direction and magnitude of variant effects. (3) ADD-ACATV: A variant-level Aggregated Cauchy Association Test (ACAT) that aggregates $P$-values, offering high sensitivity when only a small subset of variants is causal. (4) ADD-ACATO: A composite test that combines Burden, SKAT, and ACATV using weighted aggregation (e.g., equal weighting or upweighting of rare variants) to maximize power. (5) ADD-SKATO-ACAT: An extension of ADD-SKATO incorporating ACAT for enhanced power. (6) ADD-ACATV-ACAT: A composite test implemented in Regenie that integrates ACATV and ACAT. (7) ADD-BURDEN-ACAT: A burden test enhanced by ACAT, combining variant-level $P$ values to improve power under sparse alternative hypotheses. (8) ADD-BURDEN-SBAT: The Sparse Burden Association Test (SBAT), optimized for settings where multiple causal masks with consistent directional effects are present. (9) ADD-BURDEN-SBAT_NEG: A directional extension of SBAT designed to detect negative correlations between burden scores and the phenotype. (10) ADD-BURDEN-SBAT_POS: Analogous to SBAT_NEG but assumes a positive correlation between burden scores and the phenotype. Variants were grouped based on five MAF thresholds (< 5e-6, < 1e-5, < 1e-4, < 0.001, and < 0.01), and two functional masks (predicted consequence [LOF], LOF and likely deleterious missense). Regenie also estimated an 'all-mask' association strength for each genome unit, which is an aggregation of the test statistics of the individual masks. All performed tests were two-sided and the statistical significance threshold was determined to be $8.17 \times 10^{-9}$ by using a Bonferroni correction for 313 traits and 19,559 protein-coding genes.

We tested whether our signals were independent of nearby lead common variants identified in our GWAS within the same ancestry. Lead variants were lifted over to GRCh38 coordinates and mapped to the WES genotype. Variants within 1 Mb upstream or downstream of the region were considered. We re-performed the rare variant aggregate-based analyses while including the genotypes of these sentinel variants as covariates to assess conditional independence. The difference in $-\log_{10}(P\text{-values})$ before and after the conditional analysis, referred to as delta Phred, was calculated. We set an arbitrary delta threshold of 1, designating signals with a delta value below this threshold as independent from the sentinel variants.

## Colocalization with FinnGen traits

To investigate the shared genetic architecture between metabolites and clinical traits, we performed colocalization analysis using coloc R package (version 5.2.3). A total of 2179 clinical traits from the FinnGen R10 release were included[90]. For each metabolite trait, the lead variants were ranked by their statistical significance and iteratively filtered to ensure a minimum distance of 1-Mb from any variant of higher significance. For each lead variant, we extracted GWAS summary statistics for metabolites and FinnGen diseases and traits located within a 1-Mb radius as input. Priors were set to $P_1 = 1 \times 10^{-4}$, $P_2 = 1 \times 10^{-4}$, $P_{12} = 1 \times 10^{-5}$ as recommended[77]. The posterior probabilities generated by coloc indicate one of the following scenarios: no association with metabolites or disease (PP.H0); association only with metabolites (PP.H1) or disease (PP.H2); association with both, but due to different variants (PP.H3); or a shared causal variant affecting both traits (PP.H4). PP.H4 ≥ 0.8 was considered as strong evidence for colocalization.

## Mendelian randomization

For metabolite-disease pairs with colocalization evidence, we further conducted bidirectional MR analyses to investigate potential causal associations. For the forward MR analysis, we first identified potential instrumental variants for every metabolic trait based on two criteria: (1) LD $r^2 < 0.1$ within the 500 kb window, ensuring independence between variants, and (2) single-variant association $P < 5 \times 10^{-8}$, meetings genome-wide significance threshold. In the main analysis, we excluded any variants that showed significant associations ($P < 5 \times 10^{-8}$) with more than five metabolites, as these were considered at high risk of pleiotropy. To test the stability of our results, we conducted two sensitivity analyses. The first excluded variants that showed significant associations ($P < 1 \times 10^{-6}$) with more than five metabolites, while the second excluded variants that showed significant associations ($P < 5 \times 10^{-8}$) with more than three metabolites. For instruments kept in the MR analysis, we reported the exact number of associated metabolites to reflect the potential pleiotropy (Supplementary Data 31 and 32). We then harmonized the selected instrumental variables with the available GWAS summary statistics for the outcomes of interest. For the primary MR analyses, we applied the Wald ratio method for single-instrument cases and the inverse variance weighted (IVW) method when multiple instruments were available; results from additional models are presented in Supplementary Data 26–28. Findings were considered significant if FDR-corrected $Q < 0.05$ in the main analysis and $P < 0.05$ in at least one of the sensitivity analyses. For the reverse MR, where diseases were treated as the exposure and metabolites as the outcome, we applied the same clumping parameters, main analysis methods, and significance criteria as those used in the main forward MR. Analyses were performed using TwoSampleMR package (version 0.5.6).

## Reporting summary

Further information on research design is available in the Nature Portfolio Reporting Summary linked to this article.

# Data availability

The full summary statistics data have been deposited in the Figshare database at https://doi.org/10.6084/m9.figshare.29390471.v1. Individual-level genetic and phenotypic data from the UKB are available at https://biobank.ndph.ox.ac.uk/ by application. FinnGen disease GWAS is available at https://www.finngen.fi/en/access_results. The NHGRI-EBI GWAS Catalog is accessible at https://www.ebi.ac.uk/gwas/. The PubMed database can be accessed via https://pubmed.ncbi.nlm.nih.gov/. The HMDB database can be found at https://hmdb.ca/. The KEGG pathway database can be found at https://www.genome.jp/kegg/pathway.html. The PubChem database

can be found at https://pubchem.ncbi.nlm.nih.gov/. The GTEx V8 release data can be found at https://www.gtexportal.org/home/datasets. The UniProt database can be found at https://www.uniprot.org/uniprot/. The GENCODE resource can be found at https://www.gencodegenes.org/. The OMIM database can be found at https://www.omim.org/downloads. The IMPC database can be found at https://www.mousephenotype.org/data/release. The DrugBank database can be found at https://go.drugbank.com. Source data are provided with this paper.

## Code availability

The GWAS and clumping analyses were performed using PLINK (v2.0). Gene annotation was performed using bedtools (v2.31.0). LD score regression intercept, heritability, and genetic correlation effects were computed by LDSC. Single-locus level pleiotropy and the polygenicity of metabolic traits were visualized using Fuji plot (https://github.com/yk-tanigawa/fujiplot) which is based on Circos (http://circos.ca). Fine-mapping was performed using FINEMAP (v1.4.21). All other data analyses were conducted using R (version 4.2.0). R packages used for analysis and plotting include dplyr (1.1.4), data.table (1.15.4), tidyverse (2.0.0), tidyr (1.3.1), stringr (1.5.1), CJAMP (0.1.1), rtracklayer (1.58.0), TwoSampleMR (0.6.1), MRInstruments (0.3.2), coloc (5.2.3), cowplot (1.1.3), RColorBrewer (1.1.3), ggplot2 (3.5.1), ggrepel (0.9.4), and ggraph (2.1.0). ANNOVAR, http://annovar.openbioinformatics.org/en/latest/; Locuszoom, http://locuszoom.sph.umich.edu/locuszoom.

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

## Acknowledgements
This study was conducted under application number 19542 for UKB Resource. The authors gratefully thank all the participants and professionals contributing to the UKB and the FinnGen study. Ethics approval was obtained from the North West Multi-Centre Research Ethics Committee. All participants provided written informed consent. This study was funded by grants from the STI2030-Major Projects (2022ZD0211600 to J.-T.Y.), National Natural Science Foundation of China (82071201, 82271471, and 92249305 to J.-T.Y.; 82071997 to W.C.), Shanghai Municipal Science and Technology Major Project (2023SHZDZX02 to J.-T.Y. and 2018SHZDZX01 to J.-F.F.), Research Start-up Fund of Huashan Hospital (2022QD002 to J.-T.Y.), Excellence 2025 Talent Cultivation Program at Fudan University (3030277001 to J.-T.Y.), Shanghai Talent Development Funding for The Project (2019074 to J.-T.Y.), Shanghai Rising-Star Program (21QA1408700 to W.C.), 111 Project (B18015 to J.-F.F.), Humboldt Research Award (to J.-F.F.), National Key Research and Development Program of China (2023YFC3605400 to W.C.), and ZHANGJIANG LAB, Tianqiao and Chrissy Chen Institute, the State Key Laboratory of Neurobiology and Frontiers Center for Brain Science of Ministry of Education, and Shanghai Center for Brain Science and Brain-Inspired Technology, Fudan University. The funders had no involvement in the study design, data collection and analysis, decision to publish, or manuscript preparation.

## Author contributions
All authors had full access to the study data and agreed to submit it for publication. JT Yu designed the study. YX Qiang, YX Wang, XY He, YT Deng, and YJ Ge conducted the primary analyses and drafted the manuscript. JT Yu, W Cheng, JF Feng, YT Deng, YJ Ge, BS Wu and J You critically revised the manuscript, and all authors approved the final version.

## Competing interests
The authors declare no competing interests.
