## [Transparent Peer Review file · Nature Communications]

Genetic architecture of plasma metabolome in 254,825 individuals

Corresponding Author: Professor Jin-Tai Yu

Version 0:

Reviewer comments:

Reviewer #1

(Remarks to the Author)

This manuscript reports on the genetic architecture of 249 nuclear magnetic resonance-based metabolic measures and 64 derived ratios quantified in 254,825 participants of the UK Biobank (UKB). The authors identified approximately 24,000 independent SNP-trait associations based on common variants and approximately 3,000 gene-trait pairs based on aggregated rare variants. For the findings based on common variants, they investigated the pleiotropy and polygenicity, determined potential causal variants by fine-mapping, compared the findings with previously published studies, annotated potential effector genes, and performed MR and colocalization testing with regard to selected clinical traits.

Overall, the authors present a valid study design and generally use appropriate approaches to analyze the data. However, a major limitation of the study is the lack of summary statistics even for significant associations. Furthermore, comparisons with previous studies are limited to findings based on common variants, although several studies focusing on the aggregation of rare variants with respect to metabolomics have been published in the meantime. In addition, some methodological details are missing. Regarding the significance of the study, the presented showcases are not thoroughly elaborated in terms of impact and consequence of the involved variants, potential underlying mechanisms and influence on clinical traits. Therefore, the study provides few insights of biomedical relevance and reads more like a technical description of what was done without providing examples of why this might be important.

The authors are asked to address the comments below.

Major:

1. For the GWAS analysis, even for the significant associations no summary statistics are available in any of the provided supplementary tables. The authors must add the summary statistics for the significant results for metabolites and ratios to the supplementary tables and make the full summary statistics publicly available for the scientific community. In ST5, please rework the table such that each of the 24,438 independent variant-metabolite associations has their own row, to which the respective associated metabolic trait as well as all the corresponding summary statistics are added.
2. For the associations with ratios, it would be interesting to report besides the summary statistics for the ratio also those corresponding to the involved single metabolites, to assess the number of associated ratios and loci that were only identified by looking at the ratio. In the discussion in line 348f, the authors write that the GWAS of the ratios revealed variants undetectable with individual metabolites. However, since the summary statistics are not provided, the reader cannot verify this.
3. Lines 163-174: Fine-mapping showcase of XS_VLDL_TG_pct at MLXIPL:
 - a. Line 169f: "Despite the presence of seven exons at this locus, they did not exhibit a conditionally independent effect."
 - i. The distance between the 3 variants seems to be not that far. Hence, the observation of conditionally dependent signals is expectable.
 - ii. Based on Figure 5c, the reader cannot verify that the association signals of the 3 variants are conditionally dependent. Color coding of the 3 different signals or color-coding regarding LD would be helpful. Or an additional regional association plot in the supplement comparing the marginal and the conditional summary statistics.
 - iii. Figure 5c: Which summary statistics are used for this plot? The fine-mapped or marginal summary statistics? This plot is a regional association plot rather than an allelic series.
 - b. As it stands, the finemapped variants are not annotated with respect to potential functional consequences. Please incorporate more information on the SNPs such as variant impact, regulatory potential, or possible relations to gene expression or protein levels. In the beginning of this section, the authors mention multiple independent genetic mechanisms, but they did not go into detail about this for the 3 identified variants. The statement in lines 172-174 can be removed,

because without experimental validation, we don't know whether the finemapped variants are actually causal or just proxies of causal variants.

4. Lines 208-213: GWAS validation

- a. In addition to previously published studies, the authors should validate their signals in others' results from analyses of the same dataset. This will maximize overlap of metabolites, avoid annotation problems, etc. The preprint by Tambets et al analyzes the same 249 traits in the UKB and additionally combined with corresponding measures from the Estonian Biobank (<https://www.medrxiv.org/content/10.1101/2024.10.15.24315557v2.full>). Please also use this for comparison.
- b. For validating effect estimates for significant SNP-metabolite pairs identified in the current study the comparison to the corresponding SNP-metabolite pairs in previous studies: please also use a less stringent threshold (e.g. 0.05/#available SNP-metabolite pairs in the respective study) to extend the overlap between studies and enable a broader comparison.
- c. This is neither clearly mentioned in the respective section in the results nor in the methods, but Supplementary Table 11 and Supplementary Figure 6 suggest that Spearman correlation between effect estimates in this and in the previous studies were computed for each metabolite separately. Please clarify this. Supplementary Table 11 does not indicate based on how many SNPs the correlation was computed for a single metabolite. Probably, in previous studies because of the lower sample size the number of significant SNPs was limited and therefore the correlation is just based on very few SNPs for specific metabolites? Please add the number of involved SNPs to Supplementary Table 11.
- d. Line 211: For testing 1,052 correlations, the threshold $P < 0.05$ should be adjusted for multiple testing.

5. Identification of potential effector genes:

- a. Besides integrating gene expression data to identify causal genes, integrating protein levels in addition would strengthen the gene annotation approach.
- b. The current workflow assumed that common variants act through differential regulation of gene expression. It therefore misses genes where e.g., a lead coding variant may affect metabolite transport or enzyme function without modulating gene expression. Please mention this in the limitations.
- c. Please report the number of loci at which the effector gene assignment workflow did not pinpoint a candidate gene and the closest gene was used instead.

6. Rare variant associations:

- a. Please report more details about mask performance, i.e., if findings for a certain gene-trait were significant with a given mask. This could be done in an Upset plot.
- b. It would be interesting to compare your results to previous studies that focused on rare variant aggregation testing, e.g. Nag et al 2023 (NMR-based metabolomics in UKB).
- c. In addition, highlighting the complementary finding in WES data compared to GWAS findings would be interesting. There is just a single example mentioned in the discussion.
- d. Lastly, it seems that the metabolites were not transformed to force a normal distribution prior to analysis. Rare variant association analyses are sensitive to deviations of the dependent variable from a normal distribution. Supplementary Figure S10 suggests that there are traits that are still not normally distributed on the log scale. Please clarify.

7. Discussion of limitations: please discuss the limited coverage of the Nightingale platform and that the focus on lipids shapes the inferences that can be made in relation to diseases (e.g., lines 375/76), and of comparisons of e.g., heritabilities across metabolite classes. In fact, the authors should remove statements about the latter point in line 143. In general, the discussion is very generic and should better cover any new biomedically important insights gleaned from this work.

8. MR analyses: the authors have the opportunity to perform bi-directional MR analysis to look into potential reverse causality. This is important in this type of study, as for instance metabolic diseases can result in adaptations of genetically mediated metabolite processes, resulting in changes in metabolite levels. Please do so, at least for the potentially "causal" links from metabolite to trait.

9. Methods:

- a. Page 5, lines 110 ff: the authors state that clumping was done to $r^2 < 0.1$ in 10 Mb windows. However, the methods (line 499) state $r^2 < 0.01$. Which was it?
- b. The methods lack detail (e.g., about the novelty assignment. Were novel findings not previously reported for this trait? For any trait? Etc.) Points that need more detail are found below.

10. Biomedical insights: please highlight a few new findings that enable insights of biomedical relevance unknown prior to this study and discuss them in more detail. As it stands, the manuscript reads like a technical report in many parts at the cost of deeper insights into human metabolism.

Minor:

1. Line 48f: There is a huge variety of published genetic studies on the human metabolome. This could be clarified in this sentence.
2. Line 58f: There are also several published studies focusing on rare variants and the plasma metabolome. The authors could cite them here.
3. Figure 1: The authors could add a small legend explaining the color coding for the different metabolite categories present in the two top panels.
4. Line 91f: Instead of reporting the number of variant-metabolite associations without taking LD etc. between the associated variants into account, it would be more informative to report here the number of associations involving LD-independent variants as in line 112. Also in the following lines 96f, the number of associations involving LD-independent variants would be more informative. Moreover, please also mention here in the main text that variants were restricted to those of $MAF > 0.01$, and add which imputation backbone was used (this information is completely missing, also from the methods).
5. Line 102f: Also for the ratios, the number of associations involving LD-independent variants would be more informative here.
6. Figure 2: In the legend, it says each dot represents a trait-associated locus. But how are the horizontal lines defined? If each horizontal line represents a single trait, then each dot would represent a locus-trait association?
7. Line 150: Mentioning Suppl. Fig. 4 here seems not very reasonable. In addition, as explained in line 153, 3,610 is the

- number of potential causal variant-metabolite associations where the underlying variant has a $PP > 0.99$ for being causal. Hence, in Suppl. Fig.4 it should say “of 3,610 putative causal associations” instead of “of 3,610 putative causal variants”.
8. Lines 175-203: Showcase of variants associated with unsaturation degree and their influence on cholesterol measurements
 - a. Lines 179-181: Please clarify: the effect estimates for all SNPs were divided by the strongest association effect estimate with regard to all of the traits selected for Figure 5d?
 - b. Figure 5d: The names of the annotated effector genes to the selected SNPs should be added.
 - c. Line 183f: Additional information on the exonic SNP rs77960347 would be interesting, regarding its consequence, predicted damaging effects, etc. This SNPs is a missense variant and putatively damaging leading to a loss of function of the encoded protein and therefore to higher levels of the substrates.
 - d. Lines 190-199: More information on the 3 causal SNPs in FADS2 would be interesting and a specific hypothesis explaining the described association pattern (associated with increased cholesterol levels in larger VLDL particles, and decreased levels for the other traits).
 9. Line 233: There are more options how metabolites are linked to proteins, e.g. break down, excretion, conversion, transport, etc...
 10. Line 238: here and later: “filtration” does not seem the correct word for filtering.
 11. Lines 247-252: It would be informative for the reader to see the effect directions for the associations with pyruvate.
 12. Line 264: Please mention here briefly which kind of tests and masks for aggregation you used.
 13. Supplementary Table 16: Include explanation for mask1 and mask2 in the table as well as explanations for the different abbreviations for tests.
 14. Line 288f: “Furthermore, colocalization analysis confirmed shared causal variants for all 42 metabolite-disease pairs.” Is the rationale the other way round? Meaning first, colocalization analyses confirm, that disease and metabolite have shared causal variants; second, the relation between disease and metabolite is further investigated by MR analysis including the estimation of causal effects.
 15. Figure 6b: In the description of the figure it is mentioned that not only for the two ratios but also for the individual metabolites results are shown, but this is not the case
 16. Lines 313-315: “In our study, while none of these four individual metabolites showed independent links with T2D, their derived ratios demonstrated significant effects”. Because the summary statistics are not shown neither for the ratios nor for the individual metabolites the reader cannot verify this.
 17. Line 333: Here, the authors report that the fine-mapping analysis prioritized 3,610 candidate causal variants, but in line 153 they explain, that 3,610 is the number of potential causal variant-metabolite associations where the underlying variant has a $PP > 0.99$ for being causal.
 18. Methods rare variant collapsing analysis:
 - a. Please clarify which transformation of the metabolite levels were used for gene-based testing using Regenie (see above)
 - b. Please clarify how the masks were defined. Based on which annotation loss-of-function variants were determined? Based on which annotation likely deleterious missense variants were determined?
 19. Line 476: did the authors analyze gene dosages or best guess genotypes?
 20. Why did they just use FinnGen for disease data and not the UKB? How were the 83 clinical traits selected? Please provide a rationale.
 21. References: for metabolite ratios, please reference this publication: PMID: 22672667.

Reviewer #2

(Remarks to the Author)

Qiang et al have conducted a genome-wide association study of 249 circulating metabolic measures measured by NMR. They used large-scale genetic data from the UK Biobank, with a sample size $> 200,000$. They identified genome-wide significant associations at 427 loci and indicated several putative novel associations. In addition, they used whole-exome sequencing data to analyze associations with rare variants. They also characterized associations of the metabolic measure associated variants with different diseases.

The manuscript represents an important effort and is mostly clearly written. However, there are multiple studies representing overlapping approaches already available as preprints (see my specific comments). There are also some additional concerns.

1. The manuscript could be modified to address results of preprints describing overlapping approaches (Tambets et al. 2024; <https://www.medrxiv.org/content/10.1101/2024.10.15.24315557v2.full>, van der Meer et al. 2024; <https://www.medrxiv.org/content/10.1101/2024.07.30.24311254v1>); Zoodma et al. 2025; <https://www.medrxiv.org/content/10.1101/2025.01.30.25321073v1.full>). All these used UK Biobank data, and the study by Tambets et al. additionally included the Estonian Biobank, the final sample size being $> 500,000$, with more than 400,00 individuals from the UK Biobank, i.e., significantly larger than in the study by Qiang et al.
2. One of the important parts of this kind of study should be discussing the novel loci found. The authors could add multiple examples of novel loci and discuss them in relation to biology, as majority of the examples in the current version of the manuscript include well-known loci. The pyruvate associations are nicely discussed, though.
3. The authors seem to claim that there is a causal association between HDL cholesterol (or HDL cholesterol related traits?) and cardiovascular disease (they write: “We also replicated well-known associations, including those between glucose and type 2 diabetes, and HDL cholesterol and cardiovascular diseases”). Majority of research is not indicating HDL-related pathways causal for CAD, so such finding from MR analysis is unlikely to be correct and should be interpreted with caution. Could the results be confounded by pleiotropy?
4. GWAS summary statistics should be made publicly available.

Reviewer #3

(Remarks to the Author)

Reviewer #4

(Remarks to the Author)

This study is the first large-scale GWAS conducted on the most recent release of the NMR metabolite data from the UK Biobank, which includes over 275K participants. The bulk of the work consists in a series of mainstream genetic analyses following the state-of-the-art: genome-wide-association studies (GWAS) + fine-mapping + rare variants analyses + two sample Mendelian Randomization + pleiotropy analyses, etc. The work done is actually pretty extensive in its scope, and I haven't noticed any issue with any of the analyses performed. My comments are mostly about clarifying specific points and updating some display/details.

1. "A rigorous statistical framework was employed to evaluate the significance of the difference between pre- and post-statin metabolite values, incorporating adjustments for a variety of confounding factors including sex, baseline age, [...]". Please specify what the "rigorous statistical framework" is referring to. Is this a standard linear regression? What's the model used? Please provide the formula for estimating the effect and for the correction applied.
2. How are the metabolite data distributed? The authors mention "Additionally, a linear regression was established, wherein the natural log-transformed metabolites served as the dependent variables." So all metabolites were transformed, whatever their original distribution? What's the rationale?
3. Part of the manuscript focuses on GWAS of metabolite ratio. Recent work highlighted that the associations arising in ratio GWAS can be entirely denominator drive (PMID= 39818621). The authors should include a supplementary analysis showing the ratio and denominator results at top independent variants, in order to assess whether the observed ratio effect is indeed driven by a denominator effect.
4. The authors use "effector genes" on multiple occasions, but the term is not clearly defined.
5. The sample size provided should be checked and harmonized. The method states "189,846 were of white British ancestry with GWAS data, 36,445 were of non-British ancestry with GWAS data, and 197,774 were of white British ancestry with WES data." But table S1 shows British = 218,380 and Non-British = 36,445
6. In Figure 6b, the X axis label state "OR (95% CI)", however, the values for the last panel are negative. Is that the log(OR) or something else? Please confirm the scale used for all 5 panels.
7. In Figure S3, the authors may apply a hierarchical clustering on the matrix before plotting it, so that the key structure are identifiable.
8. In Figure S6 legend, specify that the Spearman correlation is between the effect estimated in the present study and effect estimates from previous studies.
9. In Figure S7, why are all the maximum values bounded at ~10-30? Is that a limitation of the method used?
10. In Figure S8, please switch the X axis label to explicit names (instead of e.g. "I9_IHD"...)

Version 1:

Reviewer comments:

Reviewer #1

(Remarks to the Author)

The authors have thoroughly revised their manuscript and addressed many of the initial comments. They shared the summary statistics of their GWAS and WES analyses, compared the associations of ratios to those of their metabolites, expanded the validation of their results compared to previous studies, included pQTLs in the workflow for assigning effector genes, clarified some of the statistical analyses in the Methods section, and expanded the Discussion section and biological interpretations.

However, several aspects require further attention and potentially clarification or modification:

Major:

1. Rare protein-coding variants:
 - o Please comment on the performance of the different tests used: ACAT-O, SKAT-O and burden tests: which test performed best? How many significant results were identified by which test? Which of these test identified the most association that were not identified with any of the other tests?
 - o Line 269f: "Of these [2,948 signals], 1,804 signals were identified using the LOF mask, 2,827 using the combined LOF and missense variant mask" -> So this means that almost all of the significant signals were detected based on the combined LOF & missense mask and that very few associations were detected exclusively by the LOF mask? Please comment on that.
 - o Line 270f: "joint tests in REGENIE, which integrate different burden masks generated from various annotation classes as well as MAF thresholds": Please clarify which burden masks (probably LOF, LOF+missense) and which MAF thresholds

(probably the ones used before) were used for the joint tests.

o Line 285f: "Among the 963 gene-trait pairs with at least one nearby common variant": why is this number smaller compared to those mentioned in line 278f? Also in ST20 there are many rows for which it says in column Q that the lead association from your GWAS is involved, but no conditional analyses were performed

o Line 289-295 comparison of rare variant aggregation testing results with Nag 2023: I was wondering about the low number of overlapping gene-metabolite pairs you mentioned in ST22, as they also used NMR-derived metabolite measurements (Nightingale). When looking at the Supplementary Table S5A of Nag et al there are definitely more significant gene-metabolite associations that overlap with your results. To assess whether your significant gene-metabolite associations were also detected by Nag and colleagues, instead of quantifying for each metabolite the number of genes for which a significant association was present in the AZ PheWAS portal, you could just check whether your significant gene-metabolite associations are present in Table S5A of Nag et al (regardless of the mask) and then report the number of gene-metabolite pairs that overlapped between your and Nags study and the number of significant gene-metabolite pairs that were not reported in Nag (although the corresponding metabolite was analyzed). Please revise accordingly.

2. Line 172f: "Despite the presence of seven exons at this locus, they did not exhibit a conditionally independent effect". This sentence is still not very insightful, as you do not infer anything from this observation. Furthermore, it is not consistent to the statement you mention in your rebuttal letter: "The putative causal variants identified by FINEMAP could be considered conditionally independent". And how did you assess the conditional dependence of these signals as you did not perform stepwise conditional analysis? As no conditional summary statistics are shown, the reader still cannot verify that the association signals of the 3 variants are conditionally dependent. Please clarify, this is confusing for readers.

3. MR: reverse causality: Based on your ST25 (filtered for sig_in_reverse_mr=="yes") and ST28, there are 80 significant MR results with reverse causality, but you did not comment on that. Could you please do so? Also reverse causality may provide interesting insights.

4. Metabolite ratios:

o Line 511: Definition of the significance threshold for the p-gain metric: the threshold of $64/(2 \times 0.05)$ for the p-gain just corrects for the number of testes ratios, but does not take into account that the ratios were tested across the whole genome. Please adjust for that as well.

o Line 106f: "Across the 64 ratios, the median proportion of lead variants that showed stronger associations with the ratio than with either individual metabolite was 33.77% (IQR: 12.74%–53.04%)." Is this proportion based on the p-gain threshold $64/(2 \times 0.05) = 640$? Please clarify this.

o Line 102-105: "Of these 64 ratios, 35 showed greater overlap with denominator-associated loci, while 29 had similar or greater overlap with numerator-associated loci. The proportion of ratio-associated loci overlapping with denominator-associated loci (0%–68.18%) and with numerator-associated loci (0%–77.78%) was comparable (Supplementary Table 6)." And in the Methods line 507: "To assess whether the ratio associations were primarily driven by the denominator metabolites⁶⁶" I assume the metabolite ratios were log-transformed (similar to the single metabolites for which the log-transformation is described in the methods), wherefore it is arbitrary which of the two metabolites is used as the numerator or the denominator (just the sign is inverse). Because of this, there is no rationale why the association of a metabolite ratio should be driven by the denominator metabolite instead of the numerator metabolite as it depends on the choice of numerator / denominator. Therefore, I think it is more important to point out in which cases testing for association with the ratio is more informative than testing for association with the single metabolites (e.g. quantified by a significant p-gain statistic)

Minor:

• In ST4, please add the sample size, for the overall association and for the male and female associations

• In ST20:

o Please add the sample size, for the overall association and for the male and female associations

o Please add a footnote in the table explaining the names of the tests mentioned in column I "TEST"

• Supplementary Tables Index Tab line 5: Typo "64 derived ratiosc"

• In ST10 in column K "Assigned to candidate gene based on", there are only two options "nearest gene" and "biological and gene expression evidence", but now the pQTL data was included as well, but this is still missing in the table

• Line 116f: Since you performed sex-stratified analyses and many of the associations seem to be sex-specific, you could comment on some interesting examples of sex-specific associations. Or, were the associations not significant in the subgroups due to power?

• Lines 175-178: You now added functional annotations to the 3 fine-mapped variants. In the rebuttal letter you wrote "We have now annotated the fine-mapped variants with predicted functional consequences using SNPnexus and the FAVOR database, and integrated eQTL and pQTL colocalization analyses to assess their potential effects on gene expression and protein levels." Where did you include your results based on colocalization analysis? If the variants have indeed a potential regulatory role, you may observe a positive association signal with eQTL data for nearby genes.

• Lines 190-198: You now added more information on the 3 causal SNPs in FADS2, but you did not comment on a hypothesis explaining the observed increased cholesterol levels in larger VLDL particles, and decreased levels for the other traits.

• Line 241: "we linked GCKR to 205 metabolites": Please mention that these assignments were just based on the nearest gene, so no biological or coloc evidence was taken into account for any of the 205 metabolite associations (as you can see in ST10, column K "Assigned to candidate gene based on").

• Discussion

o Line 337: "221 potential effector genes were assigned to 312 metabolic traits via integration of expression and biochemical evidence". You should mention here that most of these effector genes were annotated just as the nearest gene, as only for

71 variant-metabolite pairs an effector genes was annotated based on biological and colocalization evidence (as you wrote in line 236ff).

o The same issue occurs in line 364ff: "By integrating QTL colocalization and biochemical evidence, we assigned 221 unique genes to 3,610 potential causal associations." -> most of these assigned genes were just annotated as the nearest gene and not based on biological & coloc evidence.

o The same issue occurs in line 414: [our study] "yielded 785 novel putative causal signals and 221 potential effector genes" -> most of these assigned genes were just annotated as the nearest gene and not based on biological & coloc evidence.

o Line 367: example of GLUD1: "GLUD1 was assigned to 12 amino acid-related associations": mention, that for only 4 of these 12 associations the gene assignment was based on biological and coloc evidence (for the remaining 8 cases it was just the nearest gene based on ST10).

• Line 480: "Additionally, a linear regression was established, wherein the log₁₀-transformed metabolites served as the dependent variables". In the original version of the manuscript, it says that metabolite levels were natural log-transformed, which makes more sense.

Reviewer #2

(Remarks to the Author)

The manuscript has been improved after revision. The authors have added examples of biologically relevant findings, compared their findings to the studies by Tambets et al. and Nag et al., and discussed the results in more detail.

I have a minor comment: Figure 5D, right panel, shows associations of SNPs associated with the degree of unsaturation. For which allele are the effects given? Are they for the allele increasing the degree of unsaturation? This should be clarified and the results discussed accordingly.

Reviewer #3

(Remarks to the Author)

Reviewer #4

(Remarks to the Author)

I thank the author for the fair responses to my comments. I only have one minor additional comment Regarding my first point in the initial review ("A rigorous statistical framework"...), the authors provide the formula used and some explanation in the rebuttal, and state that those are include in the manuscript. However, part of this, including the formula, is absent from the manuscript. This should be included.

Version 2:

Reviewer comments:

Reviewer #1

(Remarks to the Author)

The authors have addressed most of our comment. One remaining point: why are there 2 more than twice as many associations identified with the burden test as compared to the omnibus tests? After all, the burden test should also be part of the omnibus tests, so associations detected with a burden test should also be detectable with an omnibus test

Reviewer #3

(Remarks to the Author)

The authors sincerely appreciate the critical reviews of the paper, and for the helpful way in which the reviewing editors put together a constructive list of suggestions to help improve the current work. We have now revised the paper to carefully address all the points raised. Our responses below are preceded by "-", and changes made to the paper are shown below within '...', and in red font in the revised paper.

Response to reviewer 1:

This manuscript reports on the genetic architecture of 249 nuclear magnetic resonance-based metabolic measures and 64 derived ratios quantified in 254,825 participants of the UK Biobank (UKB). The authors identified approximately 24,000 independent SNP-trait associations based on common variants and approximately 3,000 gene-trait pairs based on aggregated rare variants. For the findings based on common variants, they investigated the pleiotropy and polygenicity, determined potential causal variants by fine-mapping, compared the findings with previously published studies, annotated potential effector genes, and performed MR and colocalization testing with regard to selected clinical traits.

Overall, the authors present a valid study design and generally use appropriate approaches to analyze the data. However, a major limitation of the study is the lack of summary statistics even for significant associations. Furthermore, comparisons with previous studies are limited to findings based on common variants, although several studies focusing on the aggregation of rare variants with respect to metabolomics have been published in the meantime. In addition, some methodological details are missing. Regarding the significance of the study, the presented showcases are not thoroughly elaborated in terms of impact and consequence of the involved variants, potential underlying mechanisms and influence on clinical traits. Therefore, the study provides few insights of biomedical relevance and reads more like a technical description of what was done without providing examples of why this might be important.

The authors are asked to address the comments below.

Major:

1. For the GWAS analysis, even for the significant associations no summary statistics

are available in any of the provided supplementary tables. The authors must add the summary statistics for the significant results for metabolites and ratios to the supplementary tables and make the full summary statistics publicly available for the scientific community. In ST5, please rework the table such that each of the 24,438 independent variant-metabolite associations has their own row, to which the respective associated metabolic trait as well as all the corresponding summary statistics are added.

Response:

-Thank you for your valuable suggestion. We have made the full summary statistics for both the GWAS and WES analyses available on Figshare (<https://figshare.com/s/11feac06036d25564f79>). The reviewers can access and download these statistics immediately, and they will be publicly accessible upon principal acceptance of our manuscript. Regarding the supplementary tables, we did not include all significant associations due to their large volume (over 2.6 million significant variant-metabolite associations in the discovery analysis). However, following your suggestion, we have revised ST4 to ensure that each of the 24,438 independent variant-metabolite associations is now presented in its own row, with the corresponding metabolic trait and summary statistics included.

2. For the associations with ratios, it would be interesting to report besides the summary statistics for the ratio also those corresponding to the involved single metabolites, to assess the number of associated ratios and loci that were only identified by looking at the ratio. In the discussion in line 348f, the authors write that the GWAS of the ratios revealed variants undetectable with individual metabolites. However, since the summary statistics are not provided, the reader cannot verify this.

Response:

-Thank you for your insightful comment. In response, we have now provided, for each lead variant identified through ratio GWAS, the association P values for the ratio and its constituent metabolites (Supplementary Table 7). We have revised the Results and Methods sections to clarify these findings:

'...Across the 64 ratios, the median proportion of lead variants that showed

stronger associations with the ratio than with either individual metabolite was 33.77% (IQR: 12.74%–53.04%). Notably, a median of 21.26% (IQR: 9.53%–57.89%) of loci were uniquely identified through ratio-based analyses, underscoring their added value in uncovering novel associations beyond individual metabolites (Supplementary Table 7) ...' (Results, lines 106-110)

'...We further evaluated whether ratio associations were statistically stronger than those for individual metabolites using the P-gain metric (defined as $\frac{\min(P_{\text{numerator}}, P_{\text{denominator}})}{P_{\text{ratio}}}$, significance threshold set at $64/(2 \times 0.05) = 640$)^{13,26}. Loci uniquely identified through ratio-based analyses were determined using a Bonferroni-corrected threshold of $5 \times 10^{-8}/313 = 1.60 \times 10^{-10}$...' (Methods, lines 509-513)

3. Lines 163-174: Fine-mapping showcase of XS_VLDL_TG_pct at MLXIPL:

a. Line 169f: “Despite the presence of seven exons at this locus, they did not exhibit a conditionally independent effect.”

i. The distance between the 3 variants seems to be not that far. Hence, the observation of conditionally dependent signals is expectable.

ii. Based on Figure 5c, the reader cannot verify that the association signals of the 3 variants are conditionally dependent. Color coding of the 3 different signals or color-coding regarding LD would be helpful. Or an additional regional association plot in the supplement comparing the marginal and the conditional summary statistics.

Response:

-Thank you for your insightful comment. In this section, we performed fine-mapping using FINEMAP, which leverages LD information to identify the most likely causal configurations within a region through a shotgun stochastic search algorithm. FINEMAP can potentially identify variants with more evidence of being causal than those highlighted by a stepwise conditional analysis [1]. Therefore, the putative causal variants identified by FINEMAP could be considered conditionally independent and are more likely to have a direct effect on metabolic traits [2].

Following your suggestion, we have included a Supplementary figure 6 to display the LD structure estimated from the original (in-sample) genotype data. We have also added a clarification in the Methods to avoid confusion:

'...To pinpoint putative causal variants within these regions, we employed the FINEMAP software (version 1.4.21)²⁶, which leverages GWAS summary statistics and LD information to accurately deduce causal variants through a Bayesian variable selection approach. This approach can identify variants with more evidence of being causal than those highlighted by a stepwise conditional analysis²⁸...' (Methods, lines 531-535)

Reference:

- [1] Benner, C. et al. Prospects of Fine-Mapping Trait-Associated Genomic Regions by Using Summary Statistics from Genome-wide Association Studies. American journal of human genetics 101, 539-551 (2017). <https://doi.org:10.1016/j.ajhg.2017.08.012>
- [2] Sinnott-Armstrong N, Tanigawa Y, Amar D, et al. Genetics of 35 blood and urine biomarkers in the UK Biobank [published correction appears in Nat Genet. 2021 Nov;53(11):1622. doi: 10.1038/s41588-021-00956-2.]. Nat Genet. 2021;53(2):185-194. doi:10.1038/s41588-020-00757-z

iii. Figure 5c: Which summary statistics are used for this plot? The fine-mapped or marginal summary statistics? This plot is a regional association plot rather than an allelic series.

Response:

-Thank you for this comment. We used the summary statistics from the XS-VLDL-TG% GWAS to generate Fig. 5c. We have updated the figure legend to specify the data source and correctly identify this as a regional association plot.

b. As it stands, the finemapped variants are not annotated with respect to potential functional consequences. Please incorporate more information on the SNPs such as variant impact, regulatory potential, or possible relations to gene expression or protein levels. In the beginning of this section, the authors mention multiple independent genetic mechanisms, but they did not go into detail about this for the 3 identified

variants. The statement in lines 172-174 can be removed, because without experimental validation, we don't know whether the finemapped variants are actually causal or just proxies of causal variants.

Response:

-Thank you for this insightful comment. We have now annotated the fine-mapped variants with predicted functional consequences using SNPnexus and the FAVOR database, and integrated eQTL and pQTL colocalization analyses to assess their potential effects on gene expression and protein levels. As suggested, we have removed the statement in lines 172–174 to avoid overinterpreting the causal role without experimental evidence. For the three highlighted variants, we have expanded the discussion to include plausible mechanisms that may underlie their associations, which now reads:

'...Functional annotation using the FAVOR database²⁹ revealed that rs13234378 and rs573252567 are located in active enhancer flanking regions (EnhAF), while rs34958196 resides in a poised promoter region (PromP), suggesting potential regulatory roles for these noncoding variants...' (Results, lines 175-178)

4. Lines 208-213: GWAS validation

a. In addition to previously published studies, the authors should validate their signals in others' results from analyses of the same dataset. This will maximize overlap of metabolites, avoid annotation problems, etc. The preprint by Tambets et al analyzes the same 249 traits in the UKB and additionally combined with corresponding measures from the Estonian Biobank (<https://www.medrxiv.org/content/10.1101/2024.10.15.24315557v2.full>). Please also use this for comparison.

Response:

-As suggested, we have compared our results with three studies that include UK Biobank dataset and sixteen studies that do not. The suggested comparison with the preprint by Tambets et al. was also incorporated. The revised manuscript now states: *'...we compiled a list of 19 representative metabolomics GWAS with overlapping*

metabolites and genomic variants to our study...This comparison demonstrated substantial concordance across the datasets, *with a median Spearman's correlation of 0.98 (IQR, 0.90 to 0.99), and 1,164 of 1,327 pairwise comparisons achieving Bonferroni-corrected significance ($P < 0.05/1,327 = 3.77 \times 10^{-5}$; Supplementary Table 14, Supplementary Fig. 7). Correlations were higher for studies including UKB samples (median $R = 0.99$) than for those without (median $R = 0.93$) ...'* (Results, lines 204-212)

b. For validating effect estimates for significant SNP-metabolite pairs identified in the current study the comparison to the corresponding SNP-metabolite pairs in previous studies: please also use a less stringent threshold (e.g. $0.05/\text{\#available SNP-metabolite pairs in the respective study}$) to extend the overlap between studies and enable a broader comparison.

Response:

-Following your suggestion, we conducted sensitivity analyses using progressively less stringent significance thresholds ($P < 1.6 \times 10^{-10}$ to $P < 1.0 \times 10^{-5}$) across nine studies with full summary statistics. The validation results remained highly consistent across all thresholds, reinforcing the robustness of our findings (Figure 1). These analyses are now described in the Results section '*...Sensitivity analyses using less stringent thresholds yielded consistent results (Supplementary Table 15) ...'* (lines 212-213), and the Methods section '*...To assess robustness, we conducted sensitivity analyses using a range of significance thresholds of variants (1.00×10^{-5} to 1.60×10^{-10}) ...'* (lines 550-551).

Figure 1. Median Spearman's correlation coefficients across different significance thresholds for variants.

c. This is neither clearly mentioned in the respective section in the results nor in the methods, but Supplementary Table 11 and Supplementary Figure 6 suggest that Spearman correlation between effect estimates in this and in the previous studies were computed for each metabolite separately. Please clarify this. Supplementary Table 11 does not indicate based on how many SNPs the correlation was computed for a single metabolite. Probably, in previous studies because of the lower sample size the number of significant SNPs was limited and therefore the correlation is just based on very few SNPs for specific metabolites? Please add the number of involved SNPs to Supplementary Table 11.

Response:

-We thank the reviewer for pointing this out. We have updated Supplementary Table 14 (previously Supplementary Table 11) to include the number of variants used in the correlation analysis for each metabolite-study pair. Additionally, we have now clarified the approach in the 'Methods' section:

'...We compared effect sizes of significant variants ($P < 1.60 \times 10^{-10}$ in our study) with those reported in prior studies ($P < 5 \times 10^{-8}$ or stricter) using Spearman's rank correlation and linear regression when at least three variants overlapped. For each metabolite and study, we reported...' (Methods, lines 545-548)

d. Line 211: For testing 1,052 correlations, the threshold $P < 0.05$ should be adjusted for multiple testing.

Response:

-We have now applied Bonferroni correction for multiple testing and updated the Results section to read:

'...This comparison demonstrated substantial concordance across the datasets, *with a median Spearman's correlation of 0.98 (IQR, 0.90 to 0.99), and 1,164 of 1,327 pairwise comparisons achieving Bonferroni-corrected significance ($P < 0.05/1,327 = 3.77 \times 10^{-5}$; Supplementary Table 14, Supplementary Fig. 7) ...'*

(Results, lines 207-211)

5. Identification of potential effector genes:

a. Besides integrating gene expression data to identify causal genes, integrating protein levels in addition would strengthen the gene annotation approach.

Response:

-Thank you for this valuable suggestion. We have now integrated blood pQTL data for 4,907 proteins from a previous study [1], identifying potential effector genes for eight additional metabolite–variant pairs that were not captured by eQTL analysis alone. For another eight metabolite–variant pairs, the effector genes were supported by both pQTL and eQTL colocalization evidence, further strengthening the confidence in these gene assignments. Supplementary Tables 10 and 18, as well as Results and Methods sections have been updated to reflect this enhancement:

Reference:

[1] Ferkingstad E, Sulem P, Atlason BA, et al. Large-scale integration of the plasma proteome with genetics and disease. *Nat Genet.* 2021;53(12):1712-1721. doi:10.1038/s41588-021-00978-w

b. The current workflow assumed that common variants act through differential regulation of gene expression. It therefore misses genes where e.g., a lead coding variant may affect metabolite transport or enzyme function without modulating gene expression. Please mention this in the limitations.

Response:

-Following your suggestion, we have now addressed this limitation in the Discussion

section:

'...Furthermore, while our gene annotation workflow integrated multiple evidence sources, it primarily assumed that genetic variants influence metabolites through differential regulation of gene expression and protein abundance. This approach may not capture genes where genetic variants, directly affect protein's structure, stability, or catalytic activity...' (Discussion, lines 406-410)

c. Please report the number of loci at which the effector gene assignment workflow did not pinpoint a candidate gene and the closest gene was used instead.

Response:

-Following your suggestion, we have included this information in ST10 and the Results section.

'...Genes with both biological and colocalization evidence were designated as effector genes, leading to the annotation of 71 variant-metabolite pairs. For the remaining 3,539 pairs not annotated through this approach, the nearest genes were assigned...' (Results, lines 236-239)

6. Rare variant associations:

a. Please report more details about mask performance, i.e., if findings for a certain gene-trait were significant with a given mask. This could be done in an Upset plot.

Response:

-Thank you for your suggestion. In our rare variant association analyses, we utilized two distinct functional categories (masks): loss of function (LOF) variants and a combined mask that includes both LOF and likely deleterious missense variants. While an Upset plot may not be the most suitable for displaying the convergence of these two masks, we have provided additional clarification regarding the performance of each mask in the Results section. Specifically, we stated:

'...Of these, 1,804 signals were identified using the LOF mask, 2,827 using the combined LOF and missense variant mask, and 33 identified through joint tests in REGENIE, which integrate different burden masks generated from various

annotation classes as well as MAF thresholds...' (Results, lines 269-272)

b. It would be interesting to compare your results to previous studies that focused on rare variant aggregation testing, e.g. Nag et al 2023 (NMR-based metabolomics in UKB).

Response:

-As suggested, we compared our findings with rare variant aggregation results from Nag et al. 2023. We analyzed their publicly accessible associations ($P < 0.01$, available at <https://azphewas.com>) for validation. In the revised Results section, we now state:

'...We compared our findings with a recent WES study of metabolic traits involving 99,283 UKB participants¹⁹. Publicly available rare variant aggregation results ($P < 0.01$) from that study were analyzed. For overlapping gene-metabolite pairs (considering models with identical predicted consequences), most associations in our study were confirmed after Bonferroni correction, with validation proportions ranging from 50% to 100% (threshold defined as 0.05 divided by the number of overlapping genes per metabolite per model; Supplementary Table 22) ...' (Results, lines 289-295)

c. In addition, highlighting the complementary finding in WES data compared to GWAS findings would be interesting. There is just a single example mentioned in the discussion.

Response:

-Thank you for your valuable feedback. In response, we have revised the Results and Discussion sections to better emphasize the additional discoveries made with WES. The updated text now reads:

'...Notably, of the 2,948 gene-trait pairs identified, 1,778 had associations involving common variants in previous GWAS, 1,408 had associations involving lead variant identified in our GWAS, and the remaining 939 pairs provided new insights that complemented the GWAS findings. These results underscore the added value of WES, potentially benefiting from its deep coverage and enhanced sequencing accuracy in protein-coding genes...' (Results, lines 277-282)

'...WES additionally provided novel insights absent in GWAS. *For instance, although HMGCS2 encodes the enzyme catalyzing the first step of ketogenesis^{57,58}, our GWAS did not identify significant associations between variants in HMGCS2 and acetoacetate. In contrast, rare missense variants in HMGCS2 were significantly associated with acetoacetate in WES. Similarly, BCKDK, which encodes a kinase regulating the mitochondrial BCKDH complex involved in BCAA catabolism, was linked to isoleucine and total BCAA levels in WES but not in GWAS...*' (Discussion, lines 377-383)

d. Lastly, it seems that the metabolites were not transformed to force a normal distribution prior to analysis. Rare variant association analyses are sensitive to deviations of the dependent variable from a normal distribution. Supplementary Figure S10 suggests that there are traits that are still not normally distributed on the log scale. Please clarify.

Response:

-Thank you for your valuable comment. For our WES analysis, we applied Rank Inverse Normal Transformation (RINT) to the quantitative phenotypes using the "--apply-rint" command in REGENIE. This approach is consistent with previous studies, which have shown that log-transformation of metabolite data is not necessary when RINT is applied [1-2]. We have updated the Methods section to include this information, which now reads:

'...Rank inverse normal transformation (RINT) was applied using the '--apply-rint' command...' (Methods, lines 627-628)

Reference:

- [1] Hawkes, G. et al. Whole-genome sequencing in 333,100 individuals reveals rare non-coding single variant and aggregate associations with height. Nat Commun 15, 8549 (2024). <https://doi.org:10.1038/s41467-024-52579-w>
- [2] Gaynor, S. M. et al. Yield of genetic association signals from genomes, exomes and imputation in the UK Biobank. Nat Genet 56, 2345-2351 (2024). <https://doi.org:10.1038/s41588-024-01930-4>

7. Discussion of limitations: please discuss the limited coverage of the Nightingale platform and that the focus on lipids shapes the inferences that can be made in relation to diseases (e.g., lines 375/76), and of comparisons of e.g., heritabilities across metabolite classes. In fact, the authors should remove statements about the latter point in line 143. In general, the discussion is very generic and should better cover any new biomedically important insights gleaned from this work.

Response:

-Thank you for your suggestions. We have added discussion on the limited coverage of the Nightingale platform and its potential influence: '*...Second, NMR-based metabolomics captures fewer metabolites compared to MS, with a predominant focus on lipid-related traits. This may partly explain why most identified metabolite-disease pairs are enriched in metabolic and cardiovascular categories, and limits valid comparisons of heritability across metabolite classes...*' (Discussion, lines 401-405). Meanwhile, we have revised the statements in line 146 (previously line 143) by removing the point regarding the variable heritability across categories.

-Furthermore, we have substantially enhanced the Discussion section by incorporating several biomedically significant perspectives. Specifically, we have: (1) discussed further the implications of investigating the genetic architecture of metabolic ratios (lines 353-361), (2) elaborated on the biological validity of effector genes (lines 367-373), (3) provided new insights derived from whole exome sequencing analyses (lines 377-383), and (4) discussed the potential causal relationships between metabolites and disease pathogenesis (lines 387-398).

8. MR analyses: the authors have the opportunity to perform bi-directional MR analysis to look into potential reverse causality. This is important in this type of study, as for instance metabolic diseases can result in adaptations of genetically mediated metabolite processes, resulting in changes in metabolite levels. Please do so, at least for the potentially “causal” links from metabolite to trait.

-According to your suggestions, we have now incorporated bidirectional Mendelian

randomization analyses to comprehensively evaluate potential reverse causation between the identified metabolites and disease outcomes. The updated results and corresponding interpretation can be found in lines 302-329 and lines 386-398 of the revised manuscript.

9. Methods:

a. Page 5, lines 110 ff: the authors state that clumping was done to $r^2 < 0.1$ in 10 Mb windows. However, the methods (line 499) state $r^2 < 0.01$. Which was it?

Response:

-We apologize for the oversight. The correct clumping threshold was $r^2 < 0.01$ within 10 Mb windows, and this has now been corrected and made consistent throughout the manuscript.

b. The methods lack detail (e.g., about the novelty assignment. Were novel findings not previously reported for this trait? For any trait? Etc.) Points that need more detail are found below.

Response:

-Thank you for your valuable feedback. We have now clarified the novelty assessment methods in the manuscript as follows:

'...To ascertain whether the identified causal associations were previously documented *for the same metabolic trait*, we employed two approaches...*The remaining 785 variant-metabolic trait associations were finally recognized as novel...*' (Methods, lines 563-571)

10. Biomedical insights: please highlight a few new findings that enable insights of biomedical relevance unknown prior to this study and discuss them in more detail. As it stands, the manuscript reads like a technical report in many parts at the cost of deeper insights into human metabolism.

-We sincerely appreciate your constructive feedback. In response, we have substantially strengthened both the presentation and interpretation of our results to provide deeper mechanistic insights into human metabolism. For example, we demonstrated

how our analysis of metabolite ratios revealed genetic variants near *BCAT1* or *BCAT2* that were undetectable with individual metabolites (line 353-361). Additionally, our whole-exome sequencing provided novel insights absent in GWAS, such as associations between *BCKDK* and isoleucine (line 377-383). Meanwhile, we identified several metabolite-disease relationships that were not reported before, such as the protective role of acetate on atrial fibrillation and flutter (line 388-394).

Minor:

1. Line 48f: There is a huge variety of published genetic studies on the human metabolome. This could be clarified in this sentence.

Response:

-Following your suggestion, we have revised the relevant sentence to better reflect the context of our study:

'...Numerous studies have explored the genetic architecture of the human metabolome using diverse cohorts and analytical platforms. However, most studies faced limitations regarding either sample size or metabolite diversity⁴⁻⁹' (Introduction, lines 46-48)

2. Line 58f: There are also several published studies focusing on rare variants and the plasma metabolome. The authors could cite them here.

Response:

-Following your suggestion, we have now added relevant citations to this sentence to acknowledge previous studies focusing on rare variants.

'...while genetic studies of the metabolome have mostly focused on common variants, incorporating rare coding variants can provide novel insights¹⁹⁻²²...' (Introduction, lines 57-59)

3. Figure 1: The authors could add a small legend explaining the color coding for the different metabolite categories present in the two top panels.

Response:

-Thank you for the valuable suggestion. We have added a legend to Figure 1 to clarify the color coding of metabolite categories.

4. Line 91f: Instead of reporting the number of variant-metabolite associations without taking LD etc. between the associated variants into account, it would be more informative to report here the number of associations involving LD-independent variants as in line 112. Also in the following lines 96f, the number of associations involving LD-independent variants would be more informative. Moreover, please also mention here in the main text that variants were restricted to those of $MAF > 0.01$, and add which imputation backbone was used (this information is completely missing, also from the methods).

Response:

-Thank you for the helpful suggestion. We have revised the relevant sentences to report the number of LD-independent variant-metabolite associations. Additionally, we have explicitly stated that our GWAS analyses were restricted to common variants. Information regarding the imputation reference panels has been clarified in the Methods section.

'...The primary GWAS analysis included 189,846 white British participants and 7,924,871 common variants ($MAF > 1\%$) after quality control... In the discovery analysis, our GWAS analysis of 249 metabolic measures identified 21,132 independent variant-metabolite associations at 3,059 unique lead variants (Methods; Fig. 2). These associations were distributed across the entire NMR metabolomic spectrum, with the majority falling into the lipoprotein and lipid category, where the number of associations per trait ranged from 24 to 152. Fewer associations were observed for non-lipid traits, ranging from 7 to 85 per trait ...' (Results, lines 82-94)

'...The primary Genotyping was performed using two interconnected Affymetrix arrays: the UK BiLEVE Axiom Array for 9.9% of participants and the UKB Axiom Array for the other 90%. The Haplotype Reference Consortium (HRC)

data and UK10K haplotype resources were utilized as imputation reference panels... (Methods, lines 475-477)

5. Line 102f: Also for the ratios, the number of associations involving LD-independent variants would be more informative here.

Response:

-In line with your suggestion, we have revised the relevant sentences to:

'...We identified 3,306 independent associations for the 64 metabolite ratios (Fig. 3c). The number of associations per metabolite ratio ranged from 6 (isoleucine to valine ratio) to 122 (total cholesterol to total triglycerides ratio) ...' (Results, lines 99-102)

6. Figure 2: In the legend, it says each dot represents a trait-associated locus. But how are the horizontal lines defined? If each horizontal line represents a single trait, then each dot would represent a locus-trait association?

Response:

-Thank you for your comment. As you correctly noted, each horizontal line represents a single trait, and each dot corresponds to a locus-trait association. We have updated the figure legend accordingly.

7. Line 150: Mentioning Suppl. Fig. 4 here seems not very reasonable. In addition, as explained in line 153, 3,610 is the number of potential causal variant-metabolite associations where the underlying variant has a PP>0.99 for being causal. Hence, in Suppl. Fig.4 it should say “of 3,610 putative causal associations” instead of “of 3,610 putative causal variants”.

Response:

-Thank you for pointing this out. We have included a brief description of Supplementary Figure 4 in the Results section *'...Overall, the putative causal variants demonstrated an inverse relationship between their effect allele frequency and absolute effect size*

(Supplementary Fig.4) ...' (lines 156-158), and corrected the figure title to *'Figure S4. Absolute effect size plotted against effect allele frequency for variants in 3,610 putative causal associations'*.

8. Lines 175-203: Showcase of variants associated with unsaturation degree and their influence on cholesterol measurements

a. Lines 179-181: Please clarify: the effect estimates for all SNPs were divided by the strongest association effect estimate with regard to all of the traits selected for Figure 5d?

Response:

-Thank you for your comment. We have revised the relevant text for clarification, which now reads:

'...We standardized effect estimates to range from -1 to 1 by dividing each estimate by the strongest association effect estimate observed across all metabolic traits within each region, minimizing the influence of statistical strength and highlighting overall patterns¹⁰...' (Results, lines 182-185)

b. Figure 5d: The names of the annotated effector genes to the selected SNPs should be added.

Response:

-As suggested, we have added the annotated effector genes to the selected SNPs in Fig. 5d.

c. Line 183f: Additional information on the exonic SNP rs77960347 would be interesting, regarding its consequence, predicted damaging effects, etc. This SNPs is a missense variant and putatively damaging leading to a loss of function of the encoded protein and therefore to higher levels of the substrates.

d. Lines 190-199: More information on the 3 causal SNPs in FADS2 would be interesting and a specific hypothesis explaining the described association pattern (associated with increased cholesterol levels in larger VLDL particles, and decreased levels for the other traits).

Response:

-Thank you for your helpful suggestion. We have now revised the relevant text to incorporate additional details about these causal variants, which now reads:

'...For instance, the missense SNP rs77960347, which is putatively damaging and leads to a loss of function of LIPG, was positively associated with cholesterol levels across all lipoproteins...On the other hand, rs174575, 11:61591995_GAA_G, and rs174564 in FADS2 were associated with increased cholesterol levels in larger VLDL...These three intronic variants may exert regulatory effects, as they are located in promoter (marked by H3K4me3 and H3K9ac) and enhancer (marked by H3K4me1 and H3K27ac) regions across multiple tissues, according to the Roadmap Epigenomics project...' (Results, lines 186-195)

9. Line 233: There are more options how metabolites are linked to proteins, e.g. break down, excretion, conversion, transport, etc...

Response:

-Thank you for your insightful comment. To better reflect this complexity, we have revised the sentence to read:

'...Metabolites are not directly encoded by genes, but are linked to gene-encoded proteins through various biological processes, including synthesis, degradation, conversion, secretion, and transport...' (Results, lines 230-232)

10. Line 238: here and later: “filtration” does not seem the correct word for filtering.

Response:

-Thank you for your helpful suggestion. We have revised the relevant sentences to reflect the correct expression.

11. Lines 247-252: It would be informative for the reader to see the effect directions for the associations with pyruvate.

Response:

-As suggested, we have added information regarding the effect directions for the associations with pyruvate. The updated manuscript now reads:

'...For instance, LDHB encodes the B subunit of lactate dehydrogenase, which catalyzes the reversible conversion of pyruvate and lactate⁴³. LDHB eQTLs colocalized with both pyruvate levels and the lactate to pyruvate ratio. The variant rs138560021 emerged as the likely shared causal variant, showing negative associations with LDHB expression ($\beta = -0.33$, $P = 9.82 \times 10^{-7}$) and the lactate to pyruvate ratio ($\beta = -0.10$, $P = 1.22 \times 10^{-13}$), and positive association with pyruvate levels ($\beta = 0.09$, $P = 2.05 \times 10^{-13}$). These consistent directional effects suggest that reduced LDHB expression may disrupt pyruvate-lactate balance, in line with the enzyme's known biological function...' (Results, lines 244-251)

12. Line 264 : Please mention here briefly which kind of tests and masks for aggregation you used.

-Thank you for your suggestion. We have added the following text to the mentioned paragraph:

'...We conducted rare-variant aggregation tests using ten models based on two functional categories (Loss-of-function [LOF] variants alone or in combination with missense variants) and five MAF bins (1%, 0.1%, 0.01%, 0.001%, and 0.0005%). Three types of tests were employed: ACAT-O tests, SKAT-O tests, and burden tests. Overall, we identified 2,948 significant signals in 126 genes associated with 308 metabolic traits...' (Results, lines 263-266)

13. Supplementary Table 16 : Include explanation for mask1 and mask2 in the table as well as explanations for the different abbreviations for tests.

-Thank you for this important suggestion. Mask1 refers to loss of function (LOF) and Mask2 refers to combined LOF and likely deleterious missense variants. We have updated Supplementary Table 20 (previously Supplementary Table 16) to help readers better understand the results of WES analyses.

14. Line 288f: “Furthermore, colocalization analysis confirmed shared causal variants for all 42 metabolite-disease pairs.” Is the rationale the other way round? Meaning first, colocalization analyses confirm, that disease and metabolite have shared causal variants; second, the relation between disease and metabolite is further investigated by MR analysis including the estimation of causal effects.

Response:

-Thank you for your insightful suggestion. Following your advice, we have revised our methodological approach accordingly. The updated Results section now reads:
'...The extensive genetic correlations prompted further colocalization analysis, resulting in the identification of 72,538 metabolite-disease pairs with shared genetic determinants (Supplementary Table 24). Next, we conducted two-sample MR analyses for the colocalized pairs, resulting in the identification of 36 potentially causal associations that remained significant in sensitivity analyses and did not exhibit evidence of reverse causality...' (Results, lines 299-304)

15. Figure 6b: In the description of the figure it is mentioned that not only for the two ratios but also for the individual metabolites results are shown, but this is not the case

Response:

-We apologize for the oversight. We have now thoroughly revised Figure 6 based on the new bidirectional MR results. Both the figure and its legend have been updated to ensure consistency.

16. Lines 313-315: “In our study, while none of these four individual metabolites showed independent links with T2D, their derived ratios demonstrated significant effects”. Because the summary statistics are not shown neither for the ratios nor for the individual metabolites the reader cannot verify this.

Response:

-We appreciate your helpful comment. The summary statistics for both individual metabolites and their ratios are now available on Figshare

(<https://figshare.com/s/11feac06036d25564f79>). Following your suggestion, we conducted bidirectional MR analyses and found that associations between T2D and both ApoA1/triglycerides and glycine/tyrosine ratios remained significant in reverse MR, suggesting potential reverse causation. Consequently, we have replaced this example in the manuscript.

17. Line 333 : Here, the authors report that the fine-mapping analysis prioritized 3,610 candidate causal variants, but in line 153 they explain, that 3,610 is the number of potential causal variant-metabolite associations where the underlying variant has a $PP > 0.99$ for being causal.

-Thank you for pointing this out. We have revised the sentence to correctly state that 3,610 refers to potential causal variant-metabolite associations.

18. Methods rare variant collapsing analysis:

a. Please clarify which transformation of the metabolite levels were used for gene-based testing using Regenie (see above)

Response:

-We have updated relevant description of the rare gene-based test, which now reads:

'...Rank inverse normal transformation (RINT) was applied using the '--apply-rint' command...' (Methods, lines 627-628)

b. Please clarify how the masks were defined. Based on which annotation loss-of-function variants were determined? Based on which annotation likely deleterious missense variants were determined?

-Thank you for your suggestion. We have updated the relevant text to provide a more detailed description of how the masks were defined. The revised text now reads:

'...Rare variants (MAF < 1%) were annotated using SnpEff v4.347⁸³ against Ensembl Build 38.92, with the most severe consequence for each variant chosen across all protein coding transcripts. LOF variants were those predicted to cause frameshift insertion/deletion, splice-site alteration, stop gain, and stop loss. Likely deleterious missense variants were defined as those predicted consistently

to be deleterious by five in silico prediction tools including SIFT⁸⁴, Polyphen2_HVAR⁸⁵, LRT⁸⁶, Polyphen2_HDIV and MutationTaster⁸⁷. Variants were collapsed for each gene for the gene-based collapsing tests...' (Methods, lines 611-618)

19. Line 476: did the authors analyze gene dosages or best guess genotypes?

Response:

-The imputed data from the UK Biobank were converted from BGEN format to BED format using PLINK. The conversion process generated best-guess genotypes, selecting the genotype with the highest probability for each SNP. Subsequent GWAS analysis employed an additive linear regression model implemented in PLINK2. Accordingly, we have added the following sentence to the Methods section:

'...UKB imputed data were converted from BGEN to BED format using PLINK2.0, generating best-guess genotypes by selecting the most probable genotype per variant...' (Methods, lines 490-492)

20. Why did they just use FinnGen for disease data and not the UKB? How were the 83 clinical traits selected? Please provide a rationale.

Response:

-Thank you for your insightful comment. We used FinnGen for disease outcomes to enable a two-sample Mendelian Randomization design, wherein exposures (metabolites) and outcomes (diseases) are drawn from independent cohorts. This approach tends to yield more conservative causal estimates by reducing bias from weak instruments [1], and it has been widely adopted in high-profile GWASs published in *Nature* [2-3]. Regarding the selection of clinical traits, our initial analysis focused on 83 outcomes that were more related to metabolic processes. In this revision, we have now systematically included all 2,179 diseases and traits available from the FinnGen R10 release. These updates are reflected in the revised Results, Discussion and Methods sections.

Reference:

- [1] Burgess S, Davey Smith G, Davies NM, et al. Guidelines for performing Mendelian randomization investigations: update for summer 2023. Wellcome Open Res. 2023;4:186. Published 2023 Aug 4. doi:10.12688/wellcomeopenres.15555.3
- [2] Karjalainen MK, Karthikeyan S, Oliver-Williams C, et al. Genome-wide characterization of circulating metabolic biomarkers. Nature. 2024;628(8006):130-138. doi:10.1038/s41586-024-07148-y
- [3] Sun BB, Chiou J, Traylor M, et al. Plasma proteomic associations with genetics and health in the UK Biobank. Nature. 2023;622(7982):329-338. doi:10.1038/s41586-023-06592-6

21. References: for metabolite ratios, please reference this publication: PMID: 22672667.

Response:

- Thank you for the helpful suggestion. We have now added the recommended reference to support the analysis of metabolite ratios:

'...We further evaluated whether ratio associations were statistically stronger than those for individual metabolites using the P-gain metric (defined as $\frac{\min(P_{\text{numerator}}, P_{\text{denominator}})}{P_{\text{ratio}}}$, significance threshold set at $64/(2 \times 0.05) = 640$)^{13,67}...'

(Methods, lines 509-513)

Response to reviewer 2:

Qiang et al have conducted a genome-wide association study of 249 circulating metabolic measures measured by NMR. They used large-scale genetic data from the UK Biobank, with a sample size >200,000. They identified genome-wide significant associations at 427 loci and indicated several putative novel associations. In addition, they used whole-exome sequencing data to analyze associations with rare variants. They also characterized associations of the metabolic measure associated variants with different diseases.

The manuscript represents an important effort and is mostly clearly written. However, there are multiple studies representing overlapping approaches already available as preprints (see my specific comments). There are also some additional concerns.

1. The manuscript could be modified to address results of preprints describing overlapping approaches (Tambets et al. 2024; <https://www.medrxiv.org/content/10.1101/2024.10.15.24315557v2.full>, van der Meer et al. 2024; <https://www.medrxiv.org/content/10.1101/2024.07.30.24311254v1>); Zoodsma et al. 2025; <https://www.medrxiv.org/content/10.1101/2025.01.30.25321073v1.full>). All these used UK Biobank data, and the study by Tambets et al. additionally included the Estonian Biobank, the final sample size being > 500,000, with more than 400,00 individuals from the UK Biobank, i.e., significantly larger than in the study by Qiang et al.

Response:

-Thank you for bringing these important preprints to our attention. While these studies have leveraged UK Biobank data at scale, our work is distinguished by several unique and significant contributions. First, we incorporated 64 informative metabolite ratios along with the 249 metabolic measures provided by UK Biobank. Those biologically plausible ratios can reveal genetic variants undetectable with individual metabolites by reducing variability and reflecting pathway functionality [1], which were not investigated in these three studies. Second, we conducted whole-exome sequencing (WES) analysis, which was not performed in the study by Tambets et al. While the other two studies incorporated WES data, they relied on single statistical approaches (SKAT-O in van der Meer et al. and ACAT in Zoodsma et al.), whereas our analytical strategy employing multiple complementary tests (ACAT-O tests, SKAT-O tests, and burden tests) enhances power for detecting associated genes. Third, we performed Mendelian randomization analyses across a diverse range of diseases and traits, addressing potential horizontal pleiotropy through the systematic exclusion of pleiotropic loci at multiple thresholds. Additionally, we conducted reverse MR analyses to rule out

the possibility of reverse causation, ensuring robust causal inference. **Lastly, regarding the sample size, we would like to clarify that while Tambets et al. report >400,000 UK Biobank participants, the actual subset with both metabolomics and genetic data (~230,000 white British participants) closely matches our analytical sample.**

-We have also incorporated discussion of these relevant preprints in the Discussion section: '*...Although several recent preprints have conducted metabolomics studies using UKB data, they have not explored the genetic architecture of metabolite ratios, leaving a critical gap that our study addresses⁵²⁻⁵⁴...*' (lines 349-351), to contextualize our contributions relative to these studies.

2. One of the important parts of this kind of study should be discussing the novel loci found. The authors could add multiple examples of novel loci and discuss them in relation to biology, as majority of the examples in the current version of the manuscript include well-known loci. The pyruvate associations are nicely discussed, though.

Response:

-Thank you for this constructive feedback. In response, we have now substantially expanded our discussion of novel loci identified in our study. Specifically, we have: (1) elaborated on the biological validity of novel effector genes (lines 244-251, 367-373), (2) discussed how investigating ratios uncovers novel loci missed when analyzing individual metabolites in isolation (lines 353-361), (3) provided new insights derived from whole exome sequencing analyses (lines 377-383).

3. The authors seem to claim that there is a causal association between HDL cholesterol (or HDL cholesterol related traits?) and cardiovascular disease (they write: “We also replicated well-known associations, including those between glucose and type 2 diabetes, and HDL cholesterol and cardiovascular diseases”). Majority of research is not indicating HDL-related pathways causal for CAD, so such finding from MR analysis is unlikely to be correct and should be interpreted with caution. Could the results be confounded by pleiotropy?

Response:

-Thank you for the thoughtful comment. We agree that the causal relationships between HDL cholesterol (HDL-C) and cardiovascular diseases remain controversial in the majority of research. This inconsistency likely stems from the heterogeneity of HDL particles, with certain subfractions potentially exhibiting distinct properties [1-3]. HDL particle composition (e.g., cholesterol enrichment) and subclass distribution (e.g., large vs. small HDL) may better reflect HDL functionality (e.g., cholesterol efflux capacity, anti-inflammatory properties) than total HDL-C levels alone. In our study, the causal associations inferred from our MR analyses focused not on HDL-C levels but on the subclasses and compositions of HDL particle. We found that higher genetically predicted levels of cholesterol to total lipids percentage and cholesteryl esters to total lipids percentage in large HDL were associated with a reduced risk of ischemic heart disease (IHD). These measures reflect the cholesterol content within large HDL subclasses relative to total lipids, indicating that HDL particle composition may determine the cardioprotective function.

-Furthermore, we have applied rigorous methods to reduce the influence by pleiotropy in MR analysis. We excluded variants that showed significant associations ($P < 5 \times 10^{-8}$) with more than five metabolites in the main analysis, and confirmed robustness through stricter sensitivity analyses. MR-Egger intercepts also indicated no evidence of horizontal pleiotropy, supporting the reliability of these findings.

-In addition, we have refined the presentation of our MR findings to enhance clarity and added more discussion:

'...Furthermore, we found that higher genetically predicted levels of cholesterol to total lipids percentage and cholesteryl esters to total lipids percentage in large HDL were associated with a reduced risk of ischemic heart disease (IHD). Additionally, cholesterol to total lipids percentage in large HDL was inversely associated with the risk of coronary atherosclerosis...' (Results, lines 325-329)

'...Furthermore, we also identified the protective role of cholesterol and cholesterol esters to total lipids percentage in large HDL towards cardiovascular diseases, suggesting that HDL particle composition may contribute to its

cardioprotective function...' (Discussion, lines 394-397)

Reference:

- [1] Ouimet, M., Barrett, T. J. & Fisher, E. A. HDL and Reverse Cholesterol Transport. *Circ Res* 124, 1505-1518, doi:10.1161/circresaha.119.312617 (2019).
- [2] Pownall, H. J., Rosales, C., Gillard, B. K. & Gotto, A. M., Jr. High-density lipoproteins, reverse cholesterol transport and atherogenesis. *Nat Rev Cardiol* 18, 712-723, doi:10.1038/s41569-021-00538-z (2021).
- [3] Reyes-Soffer, G. et al. Cholesteryl Ester Transfer Protein Inhibition With Anacetrapib Decreases Fractional Clearance Rates of High-Density Lipoprotein Apolipoprotein A-I and Plasma Cholesteryl Ester Transfer Protein. *Arterioscler Thromb Vasc Biol* 36, 994-1002, doi:10.1161/atvbaha.115.306680 (2016).

4. GWAS summary statistics should be made publicly available.

-Following your suggestion, we have made the full summary statistics for both the GWAS and WES analyses available on Figshare (<https://figshare.com/s/11feac06036d25564f79>). The reviewers can access and download these statistics immediately, and they will be publicly accessible upon principal acceptance of our manuscript.

Response to reviewer 3:

Response to reviewer 4:

This study is the first large-scale GWAS conducted on the most recent release of the NMR metabolite data from the UK Biobank, which includes over 275K participants. The bulk of the work consists in a series of mainstream genetic analyses following the state-of-the-art: genome-wide-association studies (GWAS) + fine-mapping + rare

variants analyses + two sample Mendelian Randomization + pleiotropy analyses, etc. The work done is actually pretty extensive in its scope, and I haven't noticed any issue with any of the analyses performed. My comments are mostly about clarifying specific points and updating some display/details.

1. "A rigorous statistical framework was employed to evaluate the significance of the difference between pre- and post-statin metabolite values, incorporating adjustments for a variety of confounding factors including sex, baseline age, [...]". Please specify what the "rigorous statistical framework" is referring to. Is this a standard linear regression? What's the model used? Please provide the formula for estimating the effect and for the correction applied.

Response:

-Thank you for bringing this important point to our attention. We have now thoroughly revised the relevant section to provide a comprehensive description of the statistical framework used.

'For participants who initiated statin therapy between their initial enrollment (2006–2010) and their first repeat assessment (2012–2013), we assessed the impact of statin use on metabolite levels. Specifically, for each metabolic trait, we performed standard linear regression to test whether the difference between pre- and post-statin measurements was significant, adjusting for sex, baseline age, age difference between baseline and follow-up, socio-economic status (as measured by the Townsend Deprivation Index, TDI), body mass index (BMI), BMI difference between baseline and follow-up, the first 20 principal components (PCs) of genetic ancestry, and interactions between age, BMI, and sex, using the following formula:

$$\begin{aligned} y_{\text{post statin}} = & \beta_0 + \beta_1 y_{\text{pre statin}} + \beta_2 \text{sex} + \beta_3 \text{age} + \beta_4 \Delta \text{age} + \beta_5 \text{TDI} + \beta_6 \text{BMI} \\ & + \beta_7 \Delta \text{BMI} + \beta_8 \text{PC}_1 + \beta_9 \text{PC}_2 + \dots + \beta_{27} \text{PC}_{20} + \beta_{28} (\text{age} \times \text{sex}) \\ & + \beta_{29} (\text{BMI} \times \text{sex}) + \beta_{30} (\text{age} \times \text{BMI}) + \epsilon \end{aligned}$$

Only metabolites demonstrating a significant change associated with statin use ($P < 0.05$) were considered for subsequent correction. For each of these traits, we calculated the ratio of post-statin to pre-statin metabolite levels across

participants and derived the mean ratio to serve as a correction factor. Finally, this correction factor was applied to adjust the metabolite measurements of participants who were on statin therapy upon baseline assessment⁴, using the following formulas:

$$C = \frac{1}{n} \sum_{i=1}^n \frac{y_{\text{post statin},i}}{y_{\text{pre statin},i}}$$
$$y_{\text{adjusted}} = \frac{y_{\text{baseline on statin}}}{C}$$

' (Methods, lines 452-465)

Reference:

- [1] Sinnott-Armstrong, N., Tanigawa, Y., Amar, D., Mars, N., Benner, C., Aguirre, M., Venkataraman, G.R., Wainberg, M., Ollila, H.M., Kiiskinen, T., et al. (2021). Genetics of 35 blood and urine biomarkers in the UK Biobank. *Nat. Genet.* 53, 185–194. <https://doi.org/10.1038/s41588-020-00757-z>.

2. How are the metabolite data distributed? The authors mention “Additionally, a linear regression was established, wherein the natural log-transformed metabolites served as the dependent variables.” So all metabolites were transformed, whatever their original distribution? What’s the rationale?

Response:

-Thank you for your insightful comment. Log transformation is a standard approach to normalize positively skewed distributions, improving the validity of linear regression analyses. This approach is widely adopted in omics studies. For instance, the UK Biobank proteomics dataset, released by Olink, applies a \log_2 transformation to all proteins (over 2,000 in total) as part of its standard pipeline [1]. Similarly, metabolomics genetics research published in high-impact journals such as *Nature Medicine* and *Nature Genetics* also employs this practice [2–4]. In our study, 268 out of 313 metabolic traits exhibited right-skewed distributions, justifying the use of log transformation. To maintain a consistent analytical framework and facilitate comparability across all metabolites, we applied the

transformation uniformly, as per established practices.

Reference:

- [1] Sun BB, Chiou J, Traylor M, et al. Plasma proteomic associations with genetics and health in the UK Biobank. *Nature*. 2023;622(7982):329-338. doi:10.1038/s41586-023-06592-6
- [2] Surendran P, Stewart ID, Au Yeung VPW, et al. Rare and common genetic determinants of metabolic individuality and their effects on human health. *Nat Med*. 2022;28(11):2321-2332. doi:10.1038/s41591-022-02046-0
- [3] Long T, Hicks M, Yu HC, et al. Whole-genome sequencing identifies common-to-rare variants associated with human blood metabolites. *Nat Genet*. 2017;49(4):568-578. doi:10.1038/ng.3809
- [4] Chen Y, Lu T, Pettersson-Kymmer U, et al. Genomic atlas of the plasma metabolome prioritizes metabolites implicated in human diseases. *Nat Genet*. 2023;55(1):44-53. doi:10.1038/s41588-022-01270-1

3. Part of the manuscript focuses on GWAS of metabolite ratio. Recent work highlighted that the associations arising in ratio GWAS can be entirely denominator drive (PMID= 39818621). The authors should include a supplementary analysis showing the ratio and denominator results at top independent variants, in order to assess whether the observed ratio effect is indeed driven by a denominator effect.

Response:

-Thank you for your valuable suggestion. In response, we performed an additional analysis to assess whether the observed ratio associations were indeed primarily driven by denominator effects. We have updated the Results and Methods sections to discuss these findings:

'...Of these 64 ratios, 35 showed greater overlap with denominator-associated loci, while 29 had similar or greater overlap with numerator-associated loci. The proportion of ratio-associated loci overlapping with denominator-associated loci (0%–68.18%) and with numerator-associated loci (0%–77.78%) was comparable...' (Results, lines 102-105)

'...To assess whether the ratio associations were primarily driven by the denominator metabolites⁶⁶, we examined the overlap between loci associated with each ratio and those linked to its individual components...' (Methods, lines 507-509)

4. The authors use “effector genes” on multiple occasions, but the term is not clearly defined.

Response:

-Thank you for pointing this issue out. To address this, we have added a clear definition in the Methods section, which now reads:

'...Candidate genes listed in any of the queried databases as being associated with the metabolite were considered biologically relevant and were categorized as expression-relevant if they showed evidence of colocalization. Potential effector genes were subsequently defined as candidate genes that satisfied both biological relevance and expression relevance criteria. Multiple effector genes could be selected. If evidence was insufficient for all candidate genes due to limitations in current databases and QTL data, then the nearest gene was designated...' (Methods, lines 589-594)

5. The sample size provided should be checked and harmonized. The method states “189,846 were of white British ancestry with GWAS data, 36,445 were of non-British ancestry with GWAS data, and 197,774 were of white British ancestry with WES data.” But table S1 shows British = 218,380 and Non-British = 36,445

Response:

-Thank you for your careful review of our sample sizes. The total of 218,380 British individuals shown in Table S1 represents the union of two partially overlapping datasets: 189,846 individuals of white British ancestry with imputed genotype data and 197,774 individuals of white British ancestry with WES data. Many participants had both types of data available. To clarify this point, we have revised our Methods section, which now reads:

'... Among these, *218,380 were of white British ancestry (comprising 189,846 with imputed genotype data and 197,774 with WES data), and 36,445 were of non-British ancestry with imputed genotype data....*' (Methods, lines 433-435)

6. In Figure 6b, the X axis label state “OR (95% CI)”, however, the values for the last panel are negative. Is that the log(OR) or something else? Please confirm the scale used for all 5 panels.

Response:

-Thank you for pointing this out. In the updated figure, we have ensured that all panels consistently display odds ratios (ORs) with 95% confidence intervals on the X-axis.

7. In Figure S3, the authors may apply a hierarchical clustering on the matrix before plotting it, so that the key structure are identifiable.

Response:

-Thank you for your suggestion. We have updated Figure S3 by adding a hierarchical clustering tree to better illustrate the genetic correlation structure of the metabolic traits.

8. In Figure S6 legend, specify that the Spearman correlation is between the effect estimated in the present study and effect estimates from previous studies.

Response:

-Following your suggestion, we have revised the legend of Figure S7 (previously Figure S6) to read:

'Spearman's correlation coefficients between the effect estimates in the current study and the effect estimates from 19 previous studies...' (Figure legend, lines 1240-1241)

9. In Figure S7, why are all the maximum values bounded at ~10-30? Is that a limitation of the method used?

Response:

-In this figure, the $-\log_{10}(P)$ values are capped at 30 to improve readability, following practice in similar studies [1–2]. This prevents outliers from distorting the plot scale, allowing better visualization of overall data patterns. We have now clarified this in the figure legend:

'...The $-\log_{10}(P)$ values are capped at 30 to improve the figure's readability.'

(Figure legend, lines 1252-1253)

Reference:

[1] Sun, B. B. et al. Plasma proteomic associations with genetics and health in the UK Biobank. *Nature* 622, 329-338 (2023). <https://doi.org:10.1038/s41586-023-06592-6>

[2] Tabassum, R. et al. Genetic architecture of human plasma lipidome and its link to cardiovascular disease. *Nat Commun* 10, 4329 (2019). <https://doi.org:10.1038/s41467-019-11954-8>

10. In Figure S8, please switch the X axis label to explicit names (instead of e.g. “I9_IHD”...)

Response:

-Thank you for your suggestion. Our revised manuscript expanded genetic correlation analyses to 2,179 diseases, making a heatmap with explicit labels impractical. We've replaced it with a scatter plot labeling the diseases with the strongest positive or negative genetic correlations in each metabolite category.

The authors sincerely appreciate the critical reviews of the paper, and for the helpful way in which the reviewing editors put together a constructive list of suggestions to help improve the current work. We have now revised the paper to carefully address all the points raised. Our responses below are preceded by "-", and changes made to the paper are shown below within '...', and in red font in the revised paper.

Response to reviewer 1:

The authors have thoroughly revised their manuscript and addressed many of the initial comments. They shared the summary statistics of their GWAS and WES analyses, compared the associations of ratios to those of their metabolites, expanded the validation of their results compared to previous studies, included pQTLs in the workflow for assigning effector genes, clarified some of the statistical analyses in the Methods section, and expanded the Discussion section and biological interpretations. However, several aspects require further attention and potentially clarification or modification:

Major:

1. Rare protein-coding variants:

Please comment on the performance of the different tests used: ACAT-O, SKAT-O and burden tests: which test performed best? How many significant results were identified by which test? Which of these test identified the most association that were not identified with any of the other tests?

Response:

-Thank you for your constructive comments. Among the gene-based rare variant aggregation tests we performed, the burden test showed the strongest performance, identifying 2,712 significant gene-trait associations. This was followed by the composite tests ADD-SKATO-ACAT (1,169 associations) and ADD-BURDEN-ACAT (950 associations). Notably, the burden test also identified the largest number of unique associations, with 171 gene-trait pairs that were not detected by any of the other tests. We have incorporated this information in the Results section, which now reads:

'...We conducted rare-variant aggregation tests using...*ten gene-based tests including burden, ACAT, SKAT, and their respective omnibus or joint tests (Methods)*. Overall, we identified 2,948 significant gene-metabolite pairs, involving 126 genes across 308 metabolic traits, at a Bonferroni-corrected threshold of $0.05/(313 \times 19,559) = 8.17 \times 10^{-9}$...*Among all aggregation tests, burden tests identified the most associations (n = 2,712), including 171 unique to this method. ADD-SKATO-ACAT ranked second with 1,169 associations, 104 of which were unique...*' (Results, lines 269-282)

Line 269f: "Of these [2,948 signals], 1,804 signals were identified using the LOF mask, 2,827 using the combined LOF and missense variant mask" -> So this means that almost all of the significant signals were detected based on the combined LOF & missense mask and that very few associations were detected exclusively by the LOF mask? Please comment on that.

Response:

-Thank you for your insightful comments. We observed that 1,804 out of the 2,948 identified signals were detected using the LOF mask, of which 1,716 were also shared with the combined LOF and missense variant mask (Figure 1). And the combined LOF and missense variant mask identified 1,111 signals not captured by the LOF mask alone. This suggests that combining LOF and missense variants in a single mask significantly enhances detection power, aligning with previous WES studies [1-2]. Although LOF variants may exhibit larger effect sizes, the higher carrier frequency of missense variants in the population, compared to LOF variants, likely contributes to the increased detection power when both are combined. We have addressed this point in the revised manuscript, which now reads:

'...*Of these, 1,804 signals were identified using the LOF mask, 2,827 using the combined LOF and missense variant mask, and 33 were uniquely identified through joint tests in REGENIE, which integrate burden masks generated from these two annotation classes as well as all MAF thresholds. The higher carrier frequency of missense variants in the population compared to LOF variants likely contributes to the increased discovery power when both are combined...*

(Results, lines 275-280)

Figure 1. Overlapped gene-trait pairs among different masks

Reference:

- [1] Bomba L, Walter K, Guo Q, et al. Whole-exome sequencing identifies rare genetic variants associated with human plasma metabolites. *Am J Hum Genet.* 2022;109(6):1038-1054. doi:10.1016/j.ajhg.2022.04.009
- [2] He XY, Wu BS, Yang L, et al. Genetic associations of protein-coding variants in venous thromboembolism. *Nat Commun.* 2024;15(1):2819. Published 2024 Apr 1. doi:10.1038/s41467-024-47178-8

Line 270f: “joint tests in REGENIE, which integrate different burden masks generated from various annotation classes as well as MAF thresholds”: Please clarify which burden masks (probably LOF, LOF+missense) and which MAF thresholds (probably the ones used before) were used for the joint tests.

Response:

-Thank you for the suggestion. To clarify, we have revised the relevant text as follows:

'...and 33 were uniquely identified through joint tests in REGENIE, which integrate burden masks generated from these two annotation classes as well as all MAF thresholds...' (Results, lines 276-277)

'...Regenie also estimated an 'all-mask' association strength for each genome unit, which is an aggregation of the test statistics of the individual masks...' (Methods, lines 672-673)

Line 285f: “Among the 963 gene-trait pairs with at least one nearby common variant”: why is this number smaller compared to those mentioned in line 278f? Also in ST20 there are many rows for which it says in column Q that the lead association from your GWAS is involved, but no conditional analyses were performed

Response:

-Thank you for your insightful comment. The difference in numbers reflects different analytical approaches. The 1,778 and 1,408 gene-trait pairs mentioned in lines 278-279 were based on genes annotated for single common variant associations, while the 963 gene-trait pairs mentioned in line 285 represent those where nearby common variants could be directly mapped to the WES genotype for conditional analysis. We have updated the relevant text in ST21 (previously ST20) and the manuscript to clarify this:

'...examining the genes annotated for single variant associations, we showed that of the 2,948 gene-trait pairs identified, 1,778 had associations involving common variants in previous GWAS, 1,408 had associations involving lead variant identified in our GWAS, and the remaining 939 pairs provided new insights that complemented the GWAS findings...' (Results, lines 288-291)

'...Among the 963 gene-trait pairs in which at least one nearby common associated variant ($\pm 1\text{Mb}$) was directly mapped to the WES genotype, the effect sizes and P-values were not substantially attenuated in 904 pairs...' (Results, lines 286-298)

'...We tested whether our signals were independent of nearby lead common variants identified in our GWAS within the same ancestry. Lead variants were lifted over to GRCh38 coordinates and mapped to the WES genotype. Variants within 1 Mb upstream or downstream of the region were considered...' (Methods, lines 676-679)

Line 289-295 comparison of rare variant aggregation testing results with Nag 2023: I was wondering about the low number of overlapping gene-metabolite pairs you mentioned in ST22, as they also used NMR-derived metabolite measurements (Nightingale). When looking at the Supplementary Table S5A of Nag et al there are

definitely more significant gene-metabolite associations that overlap with your results. To assess whether your significant gene-metabolite associations were also detected by Nag and colleagues, instead of quantifying for each metabolite the number of genes for which a significant association was present in the AZ PheWAS portal, you could just check whether your significant gene-metabolite associations are present in Table S5A of Nag et al (regardless of the mask) and then report the number of gene-metabolite pairs that overlapped between your and Nags study and the number of significant gene-metabolite pairs that were not reported in Nag (although the corresponding metabolite was analyzed). Please revise accordingly

Response:

-According to your suggestion, we have revised the Results section to provide a more accurate and direct assessment of the overlap between our significant gene-metabolite associations and those reported by Nag et al.

'...We compared our findings with a recent WES study of metabolic traits involving 99,283 UKB participants by Nag et al.¹⁹. Among the 244 metabolites analyzed in both studies, we identified 2,524 significant gene–metabolite associations. Of these, 1,357 (53.76%) overlapped with those reported by Nag et al., while 1,167 (46.24%) were novel (Supplementary Table 23). Conversely, of the 1,476 associations reported by Nag et al., 1,357 (91.94%) were confirmed in our analysis. These results demonstrate both the robustness of our approach in replicating known associations and its power to uncover novel gene–metabolite pairs...' (Results, lines 301-307)

2. Line 172f: “Despite the presence of seven exons at this locus, they did not exhibit a conditionally independent effect”. This sentence is still not very insightful, as you do not infer anything from this observation. Furthermore, it is not consistent to the statement you mention in your rebuttal letter: “The putative causal variants identified by FINEMAP could be considered conditionally independent“. And how did you assess the conditional dependence of these signals as you did not perform stepwise conditional analysis? As no conditional summary statistics are shown, the reader still cannot verify

that the association signals of the 3 variants are conditionally dependent. Please clarify, this is confusing for readers.

Response:

-Thank you for pointing out this confusion. Our initial statement was intended to highlight that FINEMAP can deduce variants with more evidence of being causal than those highlighted by a conditional analysis. However, we agree that without performing stepwise conditional analysis, we cannot definitively assess the conditional independence of these signals. To avoid confusion, we have removed this sentence from the manuscript.

3. MR: reverse causality: Based on your ST25 (filtered for sig_in_reverse_mr=="yes") and ST28, there are 80 significant MR results with reverse causality, but you did not comment on that. Could you please do so? Also reverse causality may provide interesting insights.

Response:

-Thank you for the valuable suggestion. We have added comments on the biological interpretations of the reverse causality among metabolite-disease pairs in the Results section '*...Next, we conducted two-sample MR analyses for the colocalized pairs. Notably, reverse MR analysis revealed bidirectional causal relationships for half of the significant main MR results, including reverse causation between type 2 diabetes (T2D) and glucose...*' (lines 314-316), and the Discussion section '*...Meanwhile, the bidirectional MR analysis revealed that disease onset may lead to metabolic disturbance, providing new mechanistic insights into disease pathogenesis. For example, the bidirectional relationship between glucose and T2D suggests that maintaining glucose homeostasis may represent a key therapeutic target, while elevated glucose levels could serve as a biomarker for disease progression...*' (lines 409-413).

4. Metabolite ratios:

Line 511: Definition of the significance threshold for the p-gain metric: the threshold

of $64/(2*0.5)$ for the p-gain just corrects for the number of tested ratios, but does not take into account that the ratios were tested across the whole genome. Please adjust for that as well.

Response:

-Thank you for your thoughtful suggestion. We assume that your comment regarding the need to adjust for the “whole genome” refers to a potential correction based on the total number of variants included in the analysis. However, we chose not to correct for the number of variants in the calculation of *P*-gain for several reasons. First, we followed the *P*-gain threshold formula ($B/(2*\alpha)$, where α is the significance level and B is the number of tested metabolite pairs) as reported in the recommended literature [1], which has been shown to provide a conservative and appropriate critical value. Additionally, this *P*-gain threshold is consistent with previous studies, such as that by Shin et al. [2], which used a similar *P*-gain threshold of 250 to assess statistical evidence for associations stronger in the ratio than in individual metabolites. Second, our study included 7,924,871 variants after quality control, and applying an additional Bonferroni correction for these variants would impose an excessively stringent threshold. This could increase the risk of Type II errors (false negatives), potentially causing us to miss novel loci. Therefore, we believe our current approach strikes a reasonable balance between statistical rigor and the identification of meaningful associations.

Reference:

- [1] Petersen AK, Krumsiek J, Wägele B, et al. On the hypothesis-free testing of metabolite ratios in genome-wide and metabolome-wide association studies. *BMC Bioinformatics*. 2012;13:120. Published 2012 Jun 6. doi:10.1186/1471-2105-13-120
- [2] Shin SY, Fauman EB, Petersen AK, et al. An atlas of genetic influences on human blood metabolites. *Nat Genet*. 2014;46(6):543-550. doi:10.1038/ng.2982

Line 106f: “Across the 64 ratios, the median proportion of lead variants that showed stronger associations with the ratio than with either individual metabolite was 33.77% (IQR: 12.74%–53.04%).” Is this proportion based on the p-gain threshold $64/(2 \times 0.05)$

= 640? Please clarify this.

Line 102-105: “Of these 64 ratios, 35 showed greater overlap with denominator-associated loci, while 29 had similar or greater overlap with numerator-associated loci. The proportion of ratio-associated loci overlapping with denominator-associated loci (0%–68.18%) and with numerator-associated loci (0%–77.78%) was comparable (Supplementary Table 6).” And in the Methods line 507: “To assess whether the ratio associations were primarily driven by the denominator metabolites⁶⁶” I assume the metabolite ratios were log-transformed (similar to the single metabolites for which the log-transformation is described in the methods), wherefore it is arbitrary which of the two metabolites is used as the numerator or the denominator (just the sign is inverse). Because of this, there is no rationale why the association of a metabolite ratio should be driven by the denominator metabolite instead of the numerator metabolite as it depends on the choice of numerator / denominator. Therefore, I think it is more important to point out in which cases testing for association with the ratio is more informative than testing for association with the single metabolites (e.g. quantified by a significant p-gain statistic)

Response:

-Thank you for your constructive comment. You are correct that the proportion of lead variants showing stronger associations with the ratio than with either individual metabolite is based on the *P*-gain threshold of 640 (calculated as $64/(2 \times 0.05)$). We also agree with your important point regarding the arbitrary nature of numerator versus denominator for log-transformed ratios, making the distinction between denominator-driven and numerator-driven associations less meaningful. Following your suggestion, we have revised the Methods section to remove references to denominator-specific effects and have refocused the Results section to emphasize the added value of ratio-based analyses. The revised paragraph now reads:

'...Across the 64 ratios, ratio-associated loci showed comparable overlap with both denominator-associated (0%–68.18%) and numerator-associated loci (0%–77.78%) (Supplementary Table 6). The median proportion of lead variants that showed stronger associations with the ratio than with either individual

metabolite was 33.77% (IQR: 12.74%–53.04%), *based on a P-gain threshold of 640*. Notably, a median of 21.26% (IQR: 9.53%–57.89%) of loci were uniquely identified through ratio-based analyses, underscoring their added value in uncovering novel associations beyond individual metabolites (Supplementary Table 7) ...' (Results, lines 102-109)

Minor:

- In ST4, please add the sample size, for the overall association and for the male and female associations

Response:

-Following your suggestion, we have added the sample sizes for the overall, male, and female associations in ST4.

In ST20:

Please add the sample size, for the overall association and for the male and female associations

Response:

-As suggested, we have added the sample sizes for the overall, male, and female associations in ST21 (previously ST20).

Please add a footnote in the table explaining the names of the tests mentioned in column I “TEST”

Response:

-According to your suggestion, we have added a footnote in the table explaining the names of the tests listed in column I (“TEST”). In addition, we have clarified these test names in the Methods section of the manuscript, which now reads:

'...To account for potential heterogeneity in the proportion and direction of causal variant effects, we employed ten complementary gene-based association tests to ensure robust detection of signals⁸⁹: (1) ADD (Burden test): A conventional burden test that assumes all variants in the annotation set affect the phenotype in the same direction and with similar magnitude. (2) ADD-SKAT:

The Sequence Kernel Association Test, which allows for heterogeneity in the direction and magnitude of variant effects. (3) ADD-ACATV: A variant-level Aggregated Cauchy Association Test (ACAT) that aggregates P-values, offering high sensitivity when only a small subset of variants is causal. (4) ADD-ACATO: A composite test that combines Burden, SKAT, and ACATV using weighted aggregation (e.g., equal weighting or upweighting of rare variants) to maximize power. (5) ADD-SKATO-ACAT: An extension of ADD-SKATO incorporating ACAT for enhanced power. (6) ADD-ACATV-ACAT: A composite test implemented in Regenie that integrates ACATV and ACAT. (7) ADD-BURDEN-ACAT: A burden test enhanced by ACAT, combining variant-level P-values to improve power under sparse alternative hypotheses. (8) ADD-BURDEN-SBAT: The Sparse Burden Association Test (SBAT), optimized for settings where multiple causal masks with consistent directional effects are present. (9) ADD-BURDEN-SBAT_NEG: A directional extension of SBAT designed to detect negative correlations between burden scores and the phenotype. (10) ADD-BURDEN-SBAT_POS: Analogous to SBAT_NEG but assumes a positive correlation between burden scores and the phenotype...' (Methods, lines 652-670)

Supplementary Tables Index Tab line 5: Typo “64 derived ratiosc”

Response:

-Thank you for your careful review. We have corrected this typo in the Supplementary Tables Index.

In ST10 in column K “Assigned to candidate gene based on”, there are only two options “nearest gene” and “biological and gene expression evidence”, but now the pQTL data was included as well, but this is still missing in the table

Response:

-Thank you for pointing this out. Our gene annotation approach prioritizes genes supported by both biological evidence and eQTL/pQTL colocalization evidence; otherwise, the nearest gene is assigned. We have now updated the phrasing in ST10

to “biological and eQTL/pQTL colocalization evidence” to more accurately reflect the annotation criteria. In addition, we have provided the detailed eQTL and pQTL colocalization results in ST18.

Line 116f: Since you performed sex-stratified analyses and many of the associations seem to be sex-specific, you could comment on some interesting examples of sex-specific associations. Or, were the associations not significant in the subgroups due to power?

Response:

-Thank you for your insightful suggestion. In response, we have included an illustrative example of a sex-specific association in the revised manuscript. The variant rs768832539 was significantly associated with glycine levels in females ($P = 3.77 \times 10^{-40}$) but not in males ($P = 0.16$). Notably, this variant has also been reported to exhibit female-specific effects on sleep duration, according to the GWAS Atlas [1-2]. These findings highlight the potential for sex-specific metabolic associations to offer mechanistic insights, such as possible mediating pathways, into other important human phenotypes. Specifically, the observed sex-specific association is unlikely to result from limited power in the male subgroup, given the comparable minor allele counts in males ($n = 26,083$) and females ($n = 30,055$), with the difference unlikely to cause a substantial loss of power. We have revised the manuscript to include the following sentence:

'...Sex-specific associations were identified, exemplified by rs768832539, which was significantly associated with glycine levels in females ($P = 3.77 \times 10^{-40}$) but not in males ($P = 0.16$). Interestingly, this variant has also been reported to show female-specific effects on sleep duration according to the GWAS Atlas^{26,27}. These findings suggest that sex-specific metabolic associations may provide mechanistic insights, such as potential mediating pathways, into other important human phenotypes...' (Results, lines 118-123)

Reference:

[1] Watanabe, K., et al. A global overview of pleiotropy and genetic architecture in

complex traits. *Nat Genet* 51, 1339-1348 (2019).

[2] Lane, J.M., et al. Genome-wide association analyses of sleep disturbance traits identify new loci and highlight shared genetics with neuropsychiatric and metabolic traits. *Nat Genet* 49, 274-281 (2017).

Lines 175-178: You now added functional annotations to the 3 fine-mapped variants. In the rebuttal letter you wrote “We have now annotated the fine-mapped variants with predicted functional consequences using SNPnexus and the FAVOR database, and integrated eQTL and pQTL colocalization analyses to assess their potential effects on gene expression and protein levels.” Where did you include your results based on colocalization analysis? If the variants have indeed a potential regulatory role, you may observe a positive association signal with eQTL data for nearby genes.

Response:

-Thank you for your thoughtful comment. We have now included the results of our eQTL and pQTL colocalization analyses in ST18. For the three fine-mapped variants (rs13234378, rs573252567, and rs34958196), we did not observe strong colocalization signals with eQTL data from GTEx or pQTL data from Ferkingstad et al., respectively. This lack of significant colocalization may reflect limitations in current QTL datasets regarding tissue specificity, sample size, or resolution; or alternatively, it may indicate that these variants exert their regulatory effects through mechanisms not captured by standard QTL approaches. Despite the absence of colocalization signals, functional annotations from the FAVOR database reveal that these variants reside in regulatory regions—rs13234378 and rs573252567 are located in active enhancer flanking regions, while rs34958196 is positioned in a poised promoter region (line 179-182). This suggests that these variants may retain regulatory potential through alternative mechanisms not detectable in current QTL datasets.

Lines 190-198: You now added more information on the 3 causal SNPs in FADS2, but you did not comment on a hypothesis explaining the observed increased cholesterol

levels in larger VLDL particles, and decreased levels for the other traits.

Response:

-Thank you for your insightful comment. While we have highlighted the regulatory potential of these *FADS2* variants, the precise molecular mechanisms underlying their differential effects across lipoprotein subclasses remain to be fully elucidated through experimental efforts. We have added a brief note acknowledging this limitation:

'...The lipid profile alterations linked to these variants may explain *FADS2*'s role in cardiovascular risk, *though the underlying mechanisms for the differential effects across lipoprotein subclasses warrant further experimental investigation...*' (Results, lines 201-204)

Line 241: “we linked *GCKR* to 205 metabolites”: Please mention that these assignments were just based on the nearest gene, so no biological or coloc evidence was taken into account for any of the 205 metabolite associations (as you can see in ST10, column K “Assigned to candidate gene based on”).

Response:

-Following your suggestion, we have revised the sentence to clarify this:

'...Notably, we linked *GCKR* to 205 metabolites across diverse categories *based on genomic proximity*, suggesting its well-established role in glucokinase regulation may have far-reaching metabolic implications beyond glucose homeostasis...' (Results, lines 247-248)

Discussion

Line 337: “221 potential effector genes were assigned to 312 metabolic traits via integration of expression and biochemical evidence”. You should mention here that most of these effector genes were annotated just as the nearest gene, as only for 71 variant-metabolite pairs an effector genes was annotated based on biological and colocalization evidence (as you wrote in line 236ff).

The same issue occurs in line 364ff: “By integrating QTL colocalization and

biochemical evidence, we assigned 221 unique genes to 3,610 potential causal associations.” -> most of these assigned genes were just annotated as the nearest gene and not based on biological & coloc evidence.

The same issue occurs in line 414: [our study] “yielded 785 novel putative causal signals and 221 potential effector genes” -> most of these assigned genes were just annotated as the nearest gene and not based on biological & coloc evidence.

Response:

-Thank you for the valuable suggestion. We have now revised the corresponding text to accurately reflect the number of genes annotated via integration of expression and colocalization evidence, which now reads:

'...In addition, 45 potential effector genes were assigned to 71 variant–metabolite pairs via integration of expression and biochemical evidence...' (Discussion, lines 351-352)

'...By integrating QTL colocalization and biochemical evidence, we assigned 45 unique genes to 71 potential causal associations...' (Discussion, lines 378-379)

'...In summary, our study's substantial increase in sample size and comprehensive metabolite profiling yielded 785 novel putative causal signals and 45 potential effector genes with biological and colocalization evidence...' (Discussion, lines 431-433)

-We have also addressed this limitation in the Discussion section:

'...Furthermore, although our gene annotation workflow integrated multiple evidence sources, most genes were annotated based on genomic proximity due to current database limitations. This approach also assumed that genetic variants influence metabolites through differential regulation of gene expression and protein abundance, potentially overlooking genes where variants directly affect protein structure, stability, or catalytic activity...'

 (Discussion, lines 423-428)

Line 367: example of GLUD1: “GLUD1 was assigned to 12 amino acid-related associations”: mention, that for only 4 of these 12 associations the gene assignment was based on biological and coloc evidence (for the remaining 8 cases it was just the nearest

gene based on ST10).

Response:

-As suggested, we have now revised the relevant text to clarify this:

*'...Notably, **GLUD1** was assigned to 12 amino acid-related associations (9 novel), including those involving glutamine and its related ratios. Among these, 4 annotations were supported by biological and colocalization evidence, while the remaining were assigned based on genomic proximity...'* (Discussion, lines 380-383)

Line 480: “Additionally, a linear regression was established, wherein the log₁₀-transformed metabolites served as the dependent variables”. In the original version of the manuscript, it says that metabolite levels were natural log-transformed, which makes more sense.

Response:

-Thank you for this valuable comment. While natural log transformation is commonly used, log₁₀ transformation is also a valid method for normalizing right-skewed distributions, and has been adopted in high-profile metabolomics studies [1–2]. We have now ensured that the transformation method is correctly and consistently reported in the manuscript.

Reference:

- [1] Long T, Hicks M, Yu HC, et al. Whole-genome sequencing identifies common-to-rare variants associated with human blood metabolites. *Nat Genet.* 2017;49(4):568-578. doi:10.1038/ng.3809
- [2] Wang C, Yang C, Western D, et al. Genetic architecture of cerebrospinal fluid and brain metabolite levels and the genetic colocalization of metabolites with human traits. *Nat Genet.* 2024;56(12):2685-2695. doi:10.1038/s41588-024-01973-7

Response to reviewer 2:

The manuscript has been improved after revision. The authors have added examples of

biologically relevant findings, compared their findings to the studies by Tambets et al. and Nag et al., and discussed the results in more detail.

I have a minor comment: Figure 5D, right panel, shows associations of SNPs associated with the degree of unsaturation. For which allele are the effects given? Are they for the allele increasing the degree of unsaturation? This should be clarified and the results discussed accordingly.

Response:

-Thank you for your valuable comment. The effect sizes shown correspond to the effect alleles of each genetic variant. We have updated Figure 5D to include the alleles and revised the figure legend for clarity:

'...The effect allele for each variant is shown...' (Figure legend, line 1211)

Response to reviewer 3:

Response to reviewer 4:

I thank the author for the fair responses to my comments. I only have one minor additional comment

Regarding my first point in the initial review (“A rigorous statistical framework”...), the authors provide the formula used and some explanation in the rebuttal, and state that those are include in the manuscript. However, part of this, including the formula, is absent from the manuscript. This should be included.

Response:

-Following your suggestion. we have now included the complete formula and its accompanying explanation in the Methods section (lines 479-490). We believe this addition fully addresses your concern.

The authors sincerely appreciate the critical reviews of the paper, and for the helpful way in which the reviewing editors put together a constructive list of suggestions to help improve the current work. We have now revised the paper to carefully address all the points raised. Our responses below are preceded by "-", and changes made to the paper are shown below within '...', and in red font in the revised paper.

Response to reviewer 1:

The authors have addressed most of our comment. One remaining point: why are there 2 more than twice as many associations identified with the burden test as compared to the omnibus tests? After all, the burden test should also be part of the omnibus tests, so associations detected with a burden test should also be detectable with an omnibus test

Response:

-Thank you for your positive comments and thoughtful question. We agree with your point that the burden test is indeed a component of the omnibus tests. The burden test assumes that all variants within a gene have similar effects on the phenotype, while omnibus tests such as SKAT and ACAT allow variant effects to vary in direction and magnitude [1]. This variability in omnibus tests can reduce power when effects are consistent. Therefore, we believe that the larger number of associations detected by the burden test likely reflects that rare coding variants linked to metabolites tend to have consistent effect directions and sizes. We have now added this clarification to the manuscript:

'...The strong performance of the burden test suggests that rare coding variants associated with metabolites generally have consistent effect directions and magnitudes, aligning with the assumptions of this test...' (Results, lines 283-285)

Reference:

[1] Hawkes, G., et al. Whole-genome sequencing analysis identifies rare, large-effect noncoding variants and regulatory regions associated with circulating protein levels. Nature Genetics (2025).

Response to reviewer 3:
